# QUANTUM EXPECTATION MAXIMIZATION FOR GAUSSIAN MIXTURE MODELS

## ABSTRACT

The Expectation-Maximization (EM) algorithm is a fundamental tool in unsupervised machine learning. It is often used as an efficient way to solve Maximum Likelihood (ML) and Maximum A Posteriori estimation problems, especially for models with latent variables. It is also the algorithm of choice to fit mixture models: generative models that represent unlabelled points originating from $k$ different processes, as samples from $k$ multivariate distributions. In this work we define and use a quantum version of EM to fit a Gaussian Mixture Model. Given quantum access to a dataset of $n$ vectors of dimension $d$, our algorithm has convergence and precision guarantees similar to the classical algorithm, but the runtime is only polylogarithmic in the number of elements in the training set, and is polynomial in other parameters - as the dimension of the feature space, and the number of components in the mixture. We generalize further the algorithm by fitting any mixture model of base distributions in the exponential family. We discuss the performance of the algorithm on datasets that are expected to be classified successfully by those algorithms, arguing that on those cases we can give strong guarantees on the runtime.

## 1 INTRODUCTION

Over the last few years, the effort to find real world applications of quantum computers has greatly intensified. Along with chemistry, material sciences, finance, one of the fields where quantum computers are expected to be most beneficial is machine learning. A number of different algorithms have been proposed for quantum machine learning (Biamonte et al., 2017; Wiebe et al., 2017; Kerenidis & Prakash, 2018; Harrow et al., 2009; Subaşı et al., 2019; Farhi & Neven, 2018), both for the supervised and unsupervised setting, and despite the lack of large-scale quantum computers and quantum memory devises, some quantum algorithms have been demonstrated in proof-of-principle experiments (Li et al., 2015; Otterbach et al., 2017; Jiang et al., 2019). Here, we look at Expectation-Maximization (EM), a fundamental algorithm in unsupervised learning, that can be used to fit different mixture models and give maximum likelihood estimates with the so-called *latent variable models*. Such generative models are one of the most promising approaches for unsupervised problems. The goal of a generative model is to learn a probability distribution that is most likely to have generated the data collected in a training set $V \in \mathbb{R}^{n \times d}$ of $n$ vectors of $d$ features. Fitting the model consists in learning the parameters of a probability distribution $p$ in a certain parameterized family that best describes our vectors $v_i$. We will see that, thanks to this formulation, we can reduce a statistical problem into an optimization problem using maximum likelihood estimation (ML) estimation. The *likelihood* is the function that we use to measure how good a model is for explaining a given dataset. For a given machine learning model with parameters $\gamma$, the likelihood of our data set $V$ is the probability that the data have been generated by the model with parameters $\gamma$, assuming each point is independent and identically distributed. We think the likelihood as a function of $\gamma$, holding the dataset $V$ fixed. For $p(v_i|\gamma)$ the probability that a point $v_i$ comes from model $\gamma$, the likelihood is defined as $L(\gamma; V) := \prod_{i=1}^{n} p(v_i|\gamma)$. From this formula, we can see that in order to find the best parameters $\gamma^*$ of our model we need to solve an optimization problem. For numerical and analytical reasons, instead of maximizing the likelihood $L$, it is common practice to find the best model by maximizing the *log-likelihood* function $\ell(\gamma; V) = \log L(\gamma; V) = \sum_{i=1}^{n} \log p(v_i|\gamma)$. In this context, we want to find the model that maximizes the log-likelihood: $\gamma_{ML}^* := \arg\max_{\gamma} \sum_{i=1}^{n} \log p(v_i|\gamma)$. The procedure to calculate the log-likelihood depends on the specific model under consideration. A possible solution would be to use a gradient based optimization algorithm on $\ell$. Unfortunately, due to the indented landscape of the function, gradient based techniques often do not perform well. Therefore, it is common to solve the maximum likelihood estimation (or maximum a priori) problem using the Expectation-Maximization (EM) algorithm. EM is an iterative algorithm which is guaranteed to converge to a (local)

optimum of the likelihood. This algorithm has a striking variety of applications, and has been successfully used for medical imaging (Balafar et al., 2010), image restoration (Lagendijk et al., 1990), problems in computational biology (Fan et al., 2010), and so on. EM has been proposed in different works by different authors, but has been formalized as we know it only in 1977 (Dempster et al., 1977). For more details, we refer to (Lindsay, 1995; Bilmes et al., 1998).

In this work, we introduce Quantum Expectation-Maximization (QEM), a new algorithm for fitting mixture models. We detail its usage in the context of Gaussian Mixture Models, and we extend the result to other distributions in the exponential family. We also generalize the result by showing how to compute the MAP: the Maximum A Posteriori estimate of a mixture model. MAP estimates can be seen as the Bayesian version of maximum likelihood estimation problems. MAP estimates are often preferred over ML estimates, due to a reduced propensity to overfit. Our main result can be stated as:

**Result** (Quantum Expectation-Maximization). *(see Theorem 3.9) For a data matrix $V \in \mathbb{R}^{n \times d}$ stored in an appropriate QRAM data structure and for parameters $\delta_\theta, \delta_\mu > 0$, Quantum Expectation-Maximization (QEM) fits a Maximum Likelihood (or a Maximum A Posteriori) estimate of a Gaussian Mixture Model with $k$ components, in running time per iteration which is dominated by:*

$$\widetilde{O}\left(\frac{d^2 k^{4.5} \eta^3 \kappa^2(V) \kappa^2(\Sigma) \mu(\Sigma)}{\delta_\mu^3}\right), \tag{1}$$

where $\Sigma$ is a covariance matrix of a Gaussian distribution, $\eta$ is a parameter of the dataset related to the maximum norm of the vectors, $\delta_\theta, \delta_\mu$ are error parameters in the QEM algorithm, $\mu (< \sqrt{d})$ is a factor appearing in quantum linear algebra and $\kappa$ is the condition number of a matrix. Here we only kept the term in the running time that dominates for the range of parameters of interest. In Theorem 3.9 we explicate the running time of each step of the algorithm. The QEM algorithm runs for a number of iterations until a stopping condition is met (defined by a parameter $\epsilon_\tau > 0$) which implies a convergence to a (local) optimum.

Let's have a first high-level comparison of this result with the standard classical algorithms. The runtime of a single iteration in the standard implementation of the EM algorithm is at least $O(knd^2)$ (Pedregosa et al., 2011; Murphy, 2012). The advantage of the quantum algorithm is an exponential improvement with respect to the number of elements in the training set, albeit with a worsening on other parameters. It is crucial to find datasets where such a quantum algorithm can offer a speedup. For a reasonable range of parameters ( $d = 40$, $k = 10$, $\eta = 10$, $\delta = 0.5$, $\kappa(V) = 25$, $\kappa(\Sigma) = 5, \mu(\Sigma) = 4$) which is motivated by some experimental evidence reported in Section 4, datasets where the number of samples in the order of $O(10^{12})$ might be processed faster on a quantum computer. One should expect that some of the parameters of the quantum algorithm can be improved, especially the dependence on the condition numbers and the errors, which can make enlarge the type of datasets where QEM can offer an advantage. Note that we expect the number of iterations of the quantum algorithm to be proportional to the number of iteration of the classical case. This is to be expected since the convergence rate does not change, and it is corroborated by previous experimental evidence in a similar scenario: the number of iterations needed by q-means algorithm for convergence, is proportional to the number of iterations of the classical k-means algorithm (Kerenidis et al., 2018).

Expectation-Maximization is widely used for fitting mixture models in machine learning (Murphy, 2012). Most mixture models use a base distribution in the exponential family: Poisson (Church & Gale, 1995), Binomial, Multinomial, log-normal (Dexter & Tanner, 1972), exponential (Ghitany et al., 1994), Dirichlet multinomial (Yin & Wang, 2014), and others. EM is also used to fit mixtures of experts, mixtures of the student T distribution (which does not belong to the exponential family, and can be fitted with EM using (Liu & Rubin, 1995)) and for factor analysis, probit regression, and learning Hidden Markov Models (Murphy, 2012).

## 1.1 PREVIOUS WORK

There is a fair amount of quantum algorithms that have been proposed in the context of unsupervised learning. (Aïmeur et al., 2013; Lloyd et al., 2013b; Otterbach et al., 2017). Recently, classical machine learning algorithms were obtained by "dequantizing" quantum machine learning algorithms (Tang, 2018b;a;c; Gilyén et al., 2018a; Chia et al., 2018). The runtime of these classical algorithm is poly-logarithmic in the dimensions of the dataset. However, the polynomial dependence on the rank, the error, and the condition number, make these new algorithms impractical on interesting datasets, as shown experimentally by (Arrazola et al., 2019). Fast classical algorithm for GMM exists, albeit assuming only one shared covariance matrix (Dasgupta, 1999), and without a polylogarithmic dependence in the number of

elements in the training set. Independently of us, Miyahara, Aihara, and Lechner extended the q-means algorithm (Kerenidis et al., 2018) for Gaussian Mixture Models (Miyahara et al., 2019), using similar techniques. The main difference is that in their work the update step is performed using a hard-clustering approach (as in the $k$-means algorithm), that is for updating the centroid and the covariance matrices of a cluster $j$, only the data points for which cluster $j$ is nearest are taken into account. In our work, we use the soft clustering approach (as in the classical EM algorithm), that is for updating the centroid and the covariance matrices of cluster $j$, all the data points weighted by their responsibility for cluster $j$ are taken into account. Both approaches have merits and can offer advantages (Kearns et al., 1998).

## 2 EXPECTATION-MAXIMIZATION AND GAUSSIAN MIXTURE MODELS

As is common in machine learning literature, we introduce the Expectation-Maximization algorithm by using it to fit Gaussian Mixture Models (GMM). Mixture models are a popular generative model in machine learning. The intuition behind mixture models is to model complicated distributions by using a group of simpler (usually uni-modal) distributions. In this setting, the purpose of the learner is to model the data by fitting the joint probability distribution which most likely have generated our samples. In this section we describe GMM: probably the most used mixture model used to solve unsupervised classification problems. In fact, given a sufficiently large number of mixture components, it is possible to approximate any density defined in $\mathbb{R}^d$ (Murphy, 2012). In unsupervised settings, we are given a training set of unlabeled vectors $v_1 \cdots v_n \in \mathbb{R}^d$ which we represent as rows of a matrix $V \in \mathbb{R}^{n \times d}$. Let $y_i \in [k]$ one of the $k$ possible labels for a point $v_i$. We posit that the joint probability distribution of the data $p(v_i, y_i) = p(v_i | y_i) p(y_i)$, is defined as follow: $y_i \sim \text{Multinomial}(\theta)$ for $\theta \in \mathbb{R}^k$, and $p(v_i | y_i = j) \sim \mathcal{N}(\mu_j, \Sigma_j)$. The $\theta_j$ are the *mixing weights*, i.e. the probabilities that $y_i = j$, and $\mathcal{N}(\mu_j, \Sigma_j)$ is the Gaussian distribution centered in $\mu_j \in \mathbb{R}^d$ with covariance matrix $\Sigma_j \in \mathbb{R}^{d \times d}$. Note that the variables $y_i$ are unobserved, and thus are called *latent* variables. There is a simple interpretation for this model. We assume the data is created by first selecting an index $j \in [k]$ by sampling according to Multinomial$(\theta)$, and then a vector $v_i$ is sampled from $\mathcal{N}(\mu_j, \Sigma_j)$. Fitting a GMM to a dataset reduces to finding an assignment for the parameters: $\gamma = (\theta, \boldsymbol{\mu}, \boldsymbol{\Sigma}) = (\theta, \mu_1, \cdots, \mu_k, \Sigma_1, \cdots, \Sigma_k)$ that best maximize the log-likelihood for a given dataset. Note that the algorithm used to fit GMM can return a local minimum which might be different than $\gamma^*$: the model that represents the global optimum of the likelihood function. We use the letter $\phi$ to represent our *base distribution*, which in this case is the probability density function of a multivariate Gaussian distribution $\mathcal{N}(\mu, \Sigma)$. Using this formulation, a GMM is expressed as: $p(v) = \sum_{j=1}^{k} \theta_j \phi(v; \mu_j, \Sigma_j)$ where $\theta_j$ are the *mixing weights* of the multinomial distribution such that $\sum_{j=1}^{k} \theta_j = 1$. The probability for an observation $v_i$ to be assigned to the component $j$ is given by: $r_{ij} = \frac{\theta_j \phi(v_i; \mu_j, \Sigma_j)}{\sum_{l=1}^{k} \theta_l \phi(v_i; \mu_l, \Sigma_l)}$. This value is called *responsibility*, and corresponds to the posterior probability of the sample $i$ being assigned label $j$ by the current model. As anticipated, to find the best parameters of our generative model, we maximize the log-likelihood of the data. For GMM, the likelihood is given by the following formula (Ng, 2012):$\ell(\gamma; V) = \ell(\theta, \boldsymbol{\mu}, \boldsymbol{\Sigma}; V) = \sum_{i=1}^{n} \log p(v_i \ ; \ \theta, \boldsymbol{\mu}, \boldsymbol{\Sigma}) =$. Alas, it is seldom possible to solve maximum likelihood estimation analytically (i.e. by finding the zeroes of the derivatives of the log-like function, and this is one of those cases. Fortunately, Expectation-Maximization is an iterative algorithm that solves numerically the optimization problem of ML estimation. To complicate things, the likelihood function for GMM is not convex, and thus we might find some local minima (Hastie et al., 2009). If we were to know the latent variable $y_i$, then the log-likelihood for GMM would be:f $\ell(\gamma; V) = \sum_{i=1}^{n} \log p(v_i \mid y_i; \boldsymbol{\mu}, \boldsymbol{\Sigma}) + \log p(y_i; \theta)$ This formula can be easily maximized with respect to the parameters $\theta, \boldsymbol{\mu}$, and $\boldsymbol{\Sigma}$. In the Expectation step we calculate the missing variables $y_i$'s, given a guess of the parameters $(\theta, \boldsymbol{\mu}, \boldsymbol{\Sigma})$ of the model. Then, in the Maximization step, we use the estimate of the latent variables obtained in the Expectation step to update the estimate of the parameters. While in the Expectation step we calculate a lower bound on the likelihood, in the Maximization step we maximize it. Since at each iteration the likelihood can only increase, the algorithm is guaranteed to converge, albeit possibly to a local optimum (see (Hastie et al., 2009) for the proof). During the Expectation step all the responsibilities are calculated, while in the Maximization step we update our estimate on the parameters $\gamma^{t+1} = (\theta^{t+1}, \boldsymbol{\mu}^{t+1}, \boldsymbol{\Sigma}^{t+1})$. The stopping criterion for GMM is usually a threshold on the increment of the log-likelihood: if the log-likelihood changes less than a threshold between two iterations, then the algorithm stops. Notice that, since the value of the log-likelihood significantly depends on the amount of data points in the training sets, it is often preferable to adopt a scale-free stopping criterion, which does not depend on the number of samples. For instance, in the toolkit scikit-learn (Pedregosa et al., 2011) the stopping criterion is given by a tolerance on the average increment of the log-probability, which is chosen to be smaller than a certain $\epsilon_\tau$, say $10^{-3}$. More precisely, the stopping criterion is $|\mathbb{E}[\log p(v_i; \gamma^t)] - \mathbb{E}[\log p(v_i; \gamma^{t+1})]| < \epsilon_\tau$ which we can estimate as $|\frac{1}{n} \sum_{i=1}^{n} \log p(v_i; \gamma^t) - \frac{1}{n} \sum_{i=1}^{n} \log p(v_i; \gamma^{t+1})| < \epsilon_\tau$.

**Dataset assumptions in GMM** As in q-means (Kerenidis et al., 2018), we have an assumption on the dataset that all elements of the mixture contribute proportionally to the total responsibility: i, e

$$\frac{\sum_{i=1}^{n} r_{ij}}{\sum_{i=1}^{n} r_{il}} = \Theta(1) \quad \forall j, l \in [k]$$

This is equivalent to requiring that clusters share a comparable amount of points in the "well-clusterability" assumption in q-means (Kerenidis et al., 2018). It is also equivalent to assuming that $\theta_j/\theta_l = \Theta(1) \quad \forall j, l \in [k]$. For convenience, in this work, we also assume that the dataset is normalized such that the shortest vector has norm 1 and define $\eta := max_i \|v_i\|^2$ to be the maximum norm squared of a vector in the dataset. This is not a necessary requirement for our dataset, but it will simplify the analysis of our algorithm, allowing us to give strict bounds on the runtime.

**Preliminaries** We assume a basic understanding of quantum computing, we recommend Nielsen and Chuang (Nielsen & Chuang, 2002) for an introduction to the subject. A vector state $|v\rangle$ for $v \in \mathbb{R}^d$ is defined as $|v\rangle = \frac{1}{\|v\|} \sum_{j \in [d]} v_j |j\rangle$, where $|j\rangle$ represents $e_j$, the $j^{th}$ vector in the standard basis. The dataset is represented by a matrix $V \in \mathbb{R}^{n \times d}$, i.e. each row is a vector $v_i \in \mathbb{R}^d$ for $i \in [n]$ that represents a single data point. The cluster centers, called centroids, at time $t$ are stored in the matrix $C^t \in \mathbb{R}^{k \times d}$, such that the $j^{th}$ row $c_j^t$ for $j \in [k]$ represents the centroid of the cluster $\mathcal{C}_j^t$. We denote as $V_{\geq \tau}$ the matrix $\sum_{i=0}^{\ell} \sigma_i u_i v_i^T$ where $\sigma_\ell$ is the smallest singular value which is greater than $\tau$. With $nnz(V)$ we mean the number of non-zero elements of the rows of $V$. When we say $\kappa(V)$ we mean the condition number of the matrix $V$, that is the ratio between the biggest and the smallest (non-zero) singular value. All the tools used in this work, like quantum algorithms for computing distances and linear algebraic operations, are reported in the Supplementary Material section.

## 3 QUANTUM EXPECTATION-MAXIMIZATION FOR GMM

In this section, we present a quantum Expectation-Maximization algorithm to fit a GMM. The algorithm can also be adapted fit other mixtures models where the probability distributions belong to the exponential family. As the GMM is both intuitive and one of the most widely used mixture models, our results are presented for the GMM case.

**A robust version of the EM algorithm** Similar to the work of (Kerenidis et al., 2018), we define a $\Delta$-robust version of the EM algorithm which we use to fit a GMM. The difference between this formalization and the original EM for GMM is simple. Here we explain the *numerical error* introduced in the training algorithm.

Let $\gamma^t = (\theta^t, \boldsymbol{\mu}^t, \boldsymbol{\Sigma}^t) = (\theta^t, \mu_1^t \cdots \mu_k^t, \Sigma_1^t \cdots \Sigma_k^t)$ a model fitted by the standard EM algorithm from $\gamma^0$ an initial guess of the parameters, i.e. $\gamma^t$ is the error-free model that standard EM would have returned after $t$ iterations. Starting from the same choice of initial parameters $\gamma^0$, fitting a GMM with the QEM algorithm with $\Delta = (\delta_\theta, \delta_\mu)$ means returning a model $\overline{\gamma}^t = (\overline{\theta}^t, \overline{\boldsymbol{\mu}}^t, \overline{\boldsymbol{\Sigma}}^t)$ such that: $\left\|\overline{\theta}^t - \theta^t\right\| < \delta_\theta$, that $\left\|\overline{\mu_j}^t - \mu_j^t\right\| < \delta_\mu$ for all $j \in [k]$, and that $\left\|\overline{\Sigma_j}^t - \Sigma_j^t\right\| \leq \delta_\mu \sqrt{\eta}$.

**Quantum access to the mixture model** As in the classical algorithm, we use some subroutines to compute the responsibilities and update our current guess of the parameters. The classical algorithm has clearly two separate steps for Expectation and Maximization. In contrast, the quantum algorithm uses a subroutine to compute the responsibilities inside the step that performs the Maximization, that is the subroutines for computing responsibilities are called multiple times during the quantum Maximization step. During the quantum Maximization step, the algorithm updates the model parameters $\gamma^t$ by creating quantum states corresponding to parameters $\gamma^{t+1}$ and then recovering classical estimates for these parameters using quantum tomography or amplitude amplification. In order for this subroutines to be efficient, the values of the GMM are stored in QRAM data structures and are updated following each maximization step.

**Definition 1** (Quantum access to a GMM). *We say that we have quantum access to a GMM if the dataset $V \in \mathbb{R}^{n \times d}$ and model parameters $\theta_j \in \mathbb{R}, \mu_j \in \mathbb{R}^d, \Sigma_j \in \mathbb{R}^{d \times d}$ for all $j \in [k]$ are stored in QRAM data structures which allow us to perform in time $O(polylog(d))$ the following mappings:*

- *$|j\rangle |0\rangle \mapsto |j\rangle |\mu_j\rangle$ for all $j \in [k]$,*

- *$|j\rangle |i\rangle |0\rangle \mapsto |j\rangle |i\rangle |\sigma_i^j\rangle$ for $j \in [k], i \in [d]$ where $\sigma_i^j$ is the i-th rows of $\Sigma_j \in \mathbb{R}^{d \times d}$,*

- $|i\rangle |0\rangle \mapsto |i\rangle |v_i\rangle$ *for all* $i \in [n]$,

- $|i\rangle |0\rangle |0\rangle \mapsto |i\rangle |vec[v_i v_i^T]\rangle = |i\rangle |v_i\rangle |v_i\rangle$ *for all* $i \in [n]$,

- $|j\rangle |0\rangle \mapsto |j\rangle |\theta_j\rangle$.

---

**Algorithm 1** Quantum Expectation Maximization for GMM

---

**Require:** Quantum access to a GMM model, precision parameters $\delta_\theta, \delta_\mu$, and threshold $\epsilon_\tau$.
**Ensure:** A GMM $\overline{\gamma}^t$ that maximizes locally the likelihood $\ell(\gamma; V)$, up to tolerance $\epsilon_\tau$.

1: Use a heuristic described at the beginning of this section to determine an initial guess for $\gamma^0 = (\theta^0, \boldsymbol{\mu}^0, \boldsymbol{\Sigma}^0)$, and store these parameters in the QRAM.
2: Use Lemma 3.1 to estimate the log determinant of the matrices $\{\Sigma_j^0\}_{j=1}^k$.
3: t=0
4: **repeat**
5:     **Step 1:** Get an estimate of $\theta^{t+1}$ such that $\left\|\overline{\theta}^{t+1} - \theta^{t+1}\right\| \leq \delta_\theta$ using Lemma 3.4.
6:     **Step 2:** Get an estimate $\{\overline{\mu_j}^{t+1}\}_{j=1}^k$ by using Lemma 3.6 to estimate each $\left\|\mu_j^{t+1}\right\|$ and $|\mu_j^{t+1}\rangle$ such that $\left\|\mu_j^{t+1} - \overline{\mu_j}^{t+1}\right\| \leq \delta_\mu$.
7:     **Step 3:** Get an estimate $\{\overline{\Sigma_j}^{t+1}\}_{j=1}^k$ by using Lemma 3.7 to estimate $\left\|\Sigma_j^{t+1}\right\|_F$ and $|\Sigma_j^{t+1}\rangle$ such that $\left\|\Sigma_j^{t+1} - \overline{\Sigma_j}^{t+1}\right\| \leq \delta_\mu \sqrt{\eta}$.
8:     **Step 4:** Estimate $\mathbb{E}\overline{[p(v_i; \gamma^{t+1})]}$ up to error $\epsilon_\tau/2$ using Theorem 3.8.
9:     **Step 5:** Store $\gamma^{t+1}$ in the QRAM, and use Lemma 3.1 to estimate the determinants $\{\overline{\log det(\Sigma_j^{t+1})}\}_{j=0}^k$.
10:     $t = t + 1$
11: **until**
$$\left|\mathbb{E}\overline{[p(v_i; \gamma^t)]} - \mathbb{E}\overline{[p(v_i; \gamma^{t-1})]}\right| < \epsilon_\tau$$

12: Return $\overline{\gamma}^t = (\overline{\theta}^t, \overline{\boldsymbol{\mu}}^t, \overline{\boldsymbol{\Sigma}}^t)$

---

Quantum initialization strategies exists, and are described in the Appendix.

### 3.1 EXPECTATION

In this step of the quantum algorithm we are just showing how to compute efficiently the responsibilities as a quantum state. First, we compute the responsibilities in a quantum register, and then we show how to put them as amplitudes of a quantum state. We start by a classical algorithm used to efficiently approximate the log-determinant of the covariance matrices of the data. At each iteration of Quantum Expectation-Maximization we need to compute the determinant of the updated covariance matrices, which is done thanks to Lemma 3.1. We will see from the error analysis that in order to get an estimate of the GMM, we need to call Lemma 3.1 with precision for which the runtime of Lemma 3.1 gets subsumed by the running time of finding the updated covariance matrices through 3.7. Thus, we do not explicitly write the time to compute the determinant from now on in the algorithm and when we say that we update $\Sigma$ we include an update on the estimate of $\log(\det(\Sigma))$ as well.

**Lemma 3.1** (Determinant evaluation). *There is an algorithm that, given as input a matrix $\Sigma$ and a parameter $0 < \delta < 1$, outputs an estimate $\overline{\log(det(\Sigma))}$ such that $|\overline{\log(det(\Sigma))} - \log(det(\Sigma))| \leq \epsilon$ with probability $1 - \delta$ in time:*

$$T_{Det,\epsilon} = \widetilde{O}\left(\epsilon^{-2}\kappa(\Sigma)\log(1/\delta)nnz(\Sigma)|\log(\det(\Sigma))|\right)$$

Now we can state the main brick used to compute the responsability: a quantum algorithm for evaluating the exponent of a Gaussian distribution.

**Lemma 3.2** (Quantum Gaussian Evaluation). *Suppose we have stored in the QRAM a matrix $V \in \mathbb{R}^{n \times d}$, the centroid $\mu \in \mathbb{R}^d$ and the covariance matrix $\Sigma \in \mathbb{R}^{d \times d}$ of a multivariate Gaussian distribution $\phi(v|\mu, \Sigma)$, as well as an estimate for $\log(\det(\Sigma))$. Then for $\epsilon_1 > 0$, there exists a quantum algorithm that with probability $1 - \gamma$ performs the mapping,*

- $U_{G,\epsilon_1} : |i\rangle |0\rangle \to |i\rangle |\overline{s_i}\rangle$ such that $|s_i - \overline{s_i}| < \epsilon_1$, where $s_i = -\frac{1}{2}((v_i - \mu)^T \Sigma^{-1}(v_i - \mu) + d \log 2\pi + \log(det(\Sigma)))$ is the exponent for the Gaussian probability density function.

*The running time of the algorithm is* $T_{G,\epsilon_1} = O\left(\frac{\kappa^2(\Sigma)\mu(\Sigma)\log(1/\gamma)}{\epsilon_1}\eta\right)$.

Using controlled operations it is simple to extend the previous Theorem to work with multiple Gaussians distributions $(\mu_j, \Sigma_j)$. That is, we can control on a register $|j\rangle$ to do $|j\rangle |i\rangle |0\rangle \mapsto |j\rangle |i\rangle |\phi(v_i|\mu_j, \Sigma_j)\rangle$. In the next Lemma we will see how to obtain the responsibilities $r_{ij}$ using the previous Theorem and standard quantum circuits for doing arithmetic, controlled rotations, and amplitude amplification. The Lemma is stated in a general way, to be used with any probability distributions that belong to an exponential family.

**Lemma 3.3** (Calculating responsibilities)**.** *Suppose we have quantum access to a GMM with parameters* $\gamma^t = (\theta^t, \boldsymbol{\mu}^t, \boldsymbol{\Sigma}^t)$. *There are quantum algorithms that can:*

1. *Perform the mapping* $|i\rangle |j\rangle |0\rangle \mapsto |i\rangle |j\rangle |\overline{r_{ij}}\rangle$ *such that* $|\overline{r_{ij}} - r_{ij}| \leq \epsilon_1$ *with probability* $1 - \gamma$ *in time:*

$$T_{R_1,\epsilon_1} = \widetilde{O}(k^{1.5} \times T_{G,\epsilon_1})$$

2. *For a given* $j \in [k]$, *construct state* $|\overline{R_j}\rangle$ *such that* $\left\| |\overline{R_j}\rangle - \frac{1}{\sqrt{Z_j}} \sum_{i=0}^{n} r_{ij} |i\rangle \right\| < \epsilon_1$ *where* $Z_j = \sum_{i=0}^{n} r_{ij}^2$ *with high probability in time:*

$$T_{R_2,\epsilon_1} = \widetilde{O}(k^2 \times T_{R_1,\epsilon_1})$$

## 3.2 MAXIMIZATION

Now we need to get a new estimate for the parameters of our model. This is the idea: at each iteration we recover the new parameters of the model as quantum states, and then recover it using tomography, amplitude estimation, or sampling. Once the new model has been recovered, we update the QRAM such that we get quantum access to the model $\gamma^{t+1}$. The possibility to estimate $\theta$ comes from a call to the unitary we built to compute the responsibilities, and postselection.

**Lemma 3.4** (Computing $\theta^{t+1}$)**.** *We assume quantum access to a GMM with parameters* $\gamma^t$ *and let* $\delta_\theta > 0$ *be a precision parameter. There exists an algorithm that estimates* $\overline{\theta}^{t+1} \in \mathbb{R}^k$ *such that* $\left\| \overline{\theta}^{t+1} - \theta^{t+1} \right\| \leq \delta_\theta$ *in time*

$T_\theta = O\left(k^{3.5}\eta^{1.5}\frac{\kappa^2(\Sigma)\mu(\Sigma)}{\delta_\theta^2}\right)$

We use quantum linear algebra to transform the uniform superposition of responsibilities of the $j$-th mixture into the new centroid of the $j$-th Gaussian. Let $R_j^t \in \mathbb{R}^n$ be the vector of responsibilities for a Gaussian $j$ at iteration $t$. The following claim relates the vectors $R_j^t$ to the centroids $\mu_j^{t+1}$.

**Claim 3.5.** *Let* $R_j^t \in \mathbb{R}^n$ *be the vector of responsibilities of the points for the Gaussian* $j$ *at time* $t$, *i.e.* $(R_j^t)_i = r_{ij}^t$. *Then* $\mu_j^{t+1} \leftarrow \frac{\sum_{i=1}^{n} r_{ij}^t v_i}{\sum_{i=1}^{n} r_{ij}^t} = \frac{V^T R_j^t}{n\theta_j}$.

The proof is straightforward.

**Lemma 3.6** (Computing $\boldsymbol{\mu}_j^{t+1}$)**.** *We assume we have quantum access to a GMM with parameters* $\gamma^t$. *For a precision parameter* $\delta_\mu > 0$, *there is a quantum algorithm that calculates* $\{\overline{\mu_j}^{t+1}\}_{j=1}^k$ *such that for all* $j \in [k]$ $\left\| \overline{\mu_j}^{t+1} - \mu_j^{t+1} \right\| \leq \delta_\mu$ *in time* $T_\mu = \widetilde{O}\left(\frac{kd\eta\kappa(V)(\mu(V)+k^{3.5}\eta^{1.5}\kappa^2(\Sigma)\mu(\Sigma))}{\delta_\mu^3}\right)$

From the ability to calculate responsibility and indexing the centroids, we derive the ability to reconstruct the covariance matrix of the Gaussians as well. Again, we use quantum linear algebra subroutines and tomography to recover an approximation of each $\Sigma_j$. Recall that we have defined the matrix $V' \in \mathbb{R}^{n \times d^2}$ where the $i$-th row of $V'$ is defined as $vec[v_i v_i^T]$. For this Lemma, we assume to have the matrix stored in the QRAM. This is a reasonable assumption as the quantum states corresponding to the rows of $V'$ can be prepared as $|i\rangle |0\rangle |0\rangle \to |i\rangle |v_i\rangle |v_i\rangle$, using twice the procedure for creating the rows of $V$.

**Lemma 3.7** (Computing $\Sigma_j^{t+1}$). *We assume we have quantum access to a GMM with parameters $\gamma^t$. We also have computed estimates $\overline{\mu_j}^{t+1}$ of all centroids such that $\left\| \overline{\mu_j}^{t+1} - \mu_j^{t+1} \right\| \leq \delta_\mu$ for precision parameter $\delta_\mu > 0$. Then, there exists a quantum algorithm that outputs estimates for the new covariance matrices $\{\overline{\Sigma}_j^{t+1}\}_{j=1}^k$ such that $\left\| \Sigma_j^{t+1} - \overline{\Sigma}_j^{t+1} \right\|_F \leq \delta_\mu \sqrt{\eta}$ with high probability, in time,*

$$T_\Sigma := \widetilde{O}\Big( \frac{kd^2 \eta \kappa^2(V)(\mu(V') + \eta^2 k^{3.5}\kappa^2(\Sigma)\mu(\Sigma))}{\delta_\mu^3} \Big)$$

### 3.3 Quantum estimation of log-likelihood

Now we are going to show how it is possible to get an estimate of the log-likelihood using a quantum procedure and access to a GMM model. A good estimate is crucial, as it is used as stopping criteria for the quantum algorithm as well. Classically, we stop to iterate the EM algorithm when $|\ell(\gamma^t; V) - \ell(\gamma^{t+1}; V)| < n\epsilon$, or equivalently, we can set a tolerance on the average increase in log probability: $|\mathbb{E}[\log p(v_i; \gamma^t)] - \mathbb{E}[\log p(v_i; \gamma^{t+1})]| < \epsilon$. In the quantum algorithm it is more practical to estimate $\mathbb{E}[p(v_i; \gamma^t)] = \frac{1}{n}\sum_{i=1}^n p(v_i; \gamma)$. From this we can estimate an upper bound on the log-likelihood as $n\log\mathbb{E}[p(v_i)] = \sum_{i=1}^n \log\mathbb{E}[p(v_i)] \geq \sum_{i=1}^n \log p(v_i) = \ell(\gamma; V)$.

**Lemma 3.8** (Quantum estimation of likelihood). *We assume we have quantum access to a GMM with parameters $\gamma^t$. For $\epsilon_\tau > 0$, there exists a quantum algorithm that estimates $\mathbb{E}[p(v_i; \gamma^t)]$ with absolute error $\epsilon_\tau$ in time*

$$T_\ell = \widetilde{O}\left( k^{1.5}\eta^{1.5}\frac{\kappa^2(\Sigma)\mu(\Sigma)}{\epsilon_\tau^2} \right)$$

Putting together all the previous Lemmas, we write the main result of the work.

**Theorem 3.9** (QEM for GMM). *We assume we have quantum access to a GMM with parameters $\gamma^t$. For parameters $\delta_\theta, \delta_\mu, \epsilon_\tau > 0$, the running time of one iteration of the Quantum Expectation-Maximization (QEM) algorithm is*

$$O(T_\theta + T_\mu + T_\Sigma + T_\ell),$$

*for* $T_\theta = \widetilde{O}\left(k^{3.5}\eta^{1.5}\frac{\kappa^2(\Sigma)\mu(\Sigma)}{\delta_\theta^2}\right)$, $T_\mu = \widetilde{O}\left(\frac{kd\eta\kappa(V)(\mu(V)+k^{3.5}\eta^{1.5}\kappa^2(\Sigma)\mu(\Sigma))}{\delta_\mu^3}\right)$, $T_\Sigma = \widetilde{O}\left(\frac{kd^2\eta\kappa^2(V)(\mu(V')+\eta^2 k^{3.5}\kappa^2(\Sigma)\mu(\Sigma))}{\delta_\mu^3}\right)$ *and* $T_\ell = \widetilde{O}\left(k^{1.5}\eta^{1.5}\frac{\kappa^2(\Sigma)\mu(\Sigma)}{\epsilon_\tau^2}\right)$

*For the range of parameters of interest, the running time is dominated by $T_\Sigma$.*

The proof follows directly from the previous lemmas. Note that the cost of the whole algorithm is given by repeating the Estimation and the Maximization steps several times, until the threshold on the log-likelihood is reached. Note also that the expression of the runtime can be simplified from the observation that the cost of performing tomography on the covariance matrices $\Sigma_j$ dominates the cost.

## 4 Experimental Results

In this section, we present the results of some experiments on real datasets to bound the condition number and the other parameters of the runtime. Let's discuss the value of $\kappa(\Sigma)$, $\kappa(V)$, $\mu(\Sigma)$, and $\mu(V)$. We can thresholding the condition number by discarding small singular values of the matrix, as used in quantum linear algebra, might be advantageous. This is indeed done often in classical machine learning models, since discarding the eigenvalues smaller than a certain threshold might even improve upon the metric under consideration (i.e. often the accuracy), and is a form of regularization (Murphy, 2012, Section 6.5). This is equivalent to limiting the eccentricity of the Gaussians. We can have a similar consideration on the condition number of the dataset $\kappa(V)$. As shown before, the condition number of the matrix $V'$ appearing in Lemma 3.2 is $\kappa^2(V)$. Similarly, we can claim that the value of $\mu(V)$ will not increase significantly as we add vectors to the training set. Remember that we have some choice in picking the function $\mu$: in previous experiments we have found that choosing the maximum $\ell_1$ norm of the rows of $V$ lead to values of $\mu$ around 10, and also in this case we expect the samples of a well-clusterable (Kerenidis et al., 2018) dataset to be constant. Also, $\mu$ is bounded by the Frobenius norm of $V$. In case the matrix $V$ can be clustered with high-enough accuracy by

k-means, it has been showed that the Frobenius norm of the matrix is proportional to $\sqrt{k}$. Given that EM is a more powerful extension of k-means, we can rely on similar observations too. Usually, the number of features $d$ is much more than the number of components in the mixture, i.e. $d \gg k$, so we expect $d^2$ to dominate the $k^{3.5}$ term in the cost needed to estimate the mixing weights. This makes the runtime of a single iteration proportional to:

$$\widetilde{O}\left(\frac{d^2 k^{4.5} \eta^3 \kappa^2(V) \kappa^2(\Sigma) \mu(\Sigma)}{\delta_\mu^3}\right) \tag{2}$$

As we said, the quantum running time saves the factor that depends on the number of samples and introduces a number of other parameters. Using our experimental results we can see that when the number of samples is large enough one can expect the quantum running time to be faster than the classical one. Note as well that one can expect to save some more factors from the quantum running time with a more careful analysis.

**Experiments.** In the algorithm, we need to set the parameters $\delta_\mu$ and $\delta_\theta$ to be small enough such that the likelihood is perturbed less than $\tau/4$. We have reasons to believe that on well-clusterable data, the value of these parameters will be large enough, such as not to impact dramatically the runtime. A quantum version of k-means algorithm has already been simulated on real data under similar assumptions (Kerenidis et al., 2018). There, the authors analyzed on the MNIST dataset the performances of q-means, the $\delta$-resistant version of the classical k-means algorithm. The experiment concluded that, for datasets that are expected to be clustered nicely by this kind of clustering algorithms, the value of the parameters $\delta_\mu, \delta_\theta$ did not decrease by increasing the number of samples nor the number of features. We expect similar behaviour in the EM case, namely that for large datasets the impact on the runtime of the errors $(\delta_\mu, \delta_\theta)$ does not cancel out the exponential gain in the dependence on the number of samples. For instance, in all the experiments of q-means (Kerenidis et al., 2018) on the MNIST dataset the value of $\delta_\mu$ (which in their case was called just $\delta$) has been between $0.2$ and $0.5$. The value of $\epsilon_\tau$ is usually (for instance in scikit-learn (Pedregosa et al., 2011) ) chosen to be $10^{-3}$. The value of $\eta$ has always been below 11.

We also analyzed some other real-world dataset, which can be fitted well with the EM algorithm (Reynolds et al., 2000; APPML; Voxforge.org) to perform speaker recognition: the task of recognizing a speaker from a voice sample, having access to a training set of recorded voices of all the possible speakers. Details of the measurements are reported in the Supplementary Material section, here we report only the results in Table 1. After this, we also experimented the impact of errors on the mixing weights in the accuracy of a ML estimate of a GMM by perturbing the trained model, by adding some random noise. With a value of $\delta_\theta = 0.035$, $\delta_\mu = 0.5$ we correctly classified 98.2% utterances.

| | | $\|\Sigma\|_2$ | $|logdet(\Sigma)|$ | $\kappa^*(\Sigma)$ | $\mu(\Sigma)$ | $\mu(V)$ | $\kappa(V)$ | Acc. (%) |
|---|---|---|---|---|---|---|---|---|
| MAP | avg | 0.244 | 58.56 | 4.21 | 3.82 | 2.14 | 23.82 | 99.4 |
| | max | 2.45 | 70.08 | 50 | 4.35 | 2.79 | 40.38 | |
| ML | avg | 1.31 | 14.56 | 15.57 | 2.54 | 2.14 | 23.82 | 98.8 |
| | max | 3.44 | 92,3 | 50 | 3.67 | 2.79 | 40.38 | |

Table 1: We estimate some of the parameters of the VoxForge (Voxforge.org) dataset. The averages for the matrix $V$ are taken over 34 samples, while for $\Sigma$ is over 170 samples. The accuracy reported in the experiments is measured on 170 samples in the test set, after the threshold on the eigenvalues of $\Sigma$. Each model is the result of the best of 3 different initializations of the EM algorithm. The first and the second column are the maximum singular values of all the covariance matrices, and the absolute value of the log-determinant. The column $\kappa^*(\Sigma)$ consist in the thresholded condition number for the covariance matrices.

In conclusion, the experimental results suggest that the influence of the extra parameters in the quantum running time (condition thresholds, errors, etc.) is moderate. This allows us to be optimistic that, when quantum computers with quantum access to data become a reality, our algorithm (and improved versions that reduce even more the complexity with respect to these extra parameters) could be useful in analyzing large datasets.

## 5   ACKNOWLEDGEMENTS

We would like to thank the authors of (Miyahara et al., 2019) for sharing an early version of their manuscript with us. Part of this research was supported by the projects: ANR QuDATA and QuantERA QuantAlgo.

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

# 6 SUPPLEMENTARY MATERIAL

## 6.1 PREVIOUS RESULTS USED IN THE PROOFS

Here we report useful Theorems, Lemmas, and Claims that we use in the main text.

**Theorem 6.1** (Determinant estimation (Boutsidis et al., 2017))**.** *Let $M \in \mathbb{R}^{d \times d}$ be a positive definite matrix with eigenvalues in the interval $(\sigma_{min}, 1)$. Then for all $\delta \in (0, 1)$ and $\epsilon > 0$ there is a classical algorithm that outputs $\overline{\log(det(M))}$ such that $|\overline{\log(det(M))} - \log(det(M))| \leq 2\epsilon |\log(det(M))|$ with probability at least $1 - \delta$ in time*

$$T_{Det,\epsilon} := O\left(\frac{\log(1/\epsilon)\log(1/\delta)}{\epsilon^2 \sigma_{min}} nnz(M)\right).$$

**Theorem 6.2** (Multivariate Mean Value Theorem (Rudin et al., 1964))**.** *Let $U$ be an open set of $\mathbb{R}^d$. For a differentiable functions $f : U \mapsto \mathbb{R}$ it holds that $\forall x, y \in U$, $\exists c$ such that $f(x) - f(y) = \nabla f(c) \cdot (x - y)$.*

**Theorem 6.3** (Componentwise *Softmax* function $\sigma_j(v)$ is Lipschitz continuous)**.** *For $d > 2$, let $\sigma_j : \mathbb{R}^d \mapsto (0, 1)$ be the* softmax *function defined as $\sigma_j(v) = \frac{e^{v_j}}{\sum_{l=1}^d e^{v_l}}$ Then $\sigma_j$ is Lipschitz continuous, with $K \leq \sqrt{2}$.*

---

**Algorithm 2** Expectation-Maximization for GMM

---

**Require:** Dataset $V$, tolerance $\tau > 0$.
**Ensure:** A GMM $\gamma^t = (\theta^t, \boldsymbol{\mu}^t, \boldsymbol{\Sigma}^t)$ that maximizes locally the likelihood $\ell(\gamma; V)$ up to tolerance $\tau$.

1: Select $\gamma^0 = (\theta^0, \boldsymbol{\mu}^0, \boldsymbol{\Sigma}^0)$ using classical initialization strategies described in Subsection 6.3.
2: $t = 0$
3: **repeat**
4:     **Expectation**
    $\forall i, j$, calculate the responsibilities as:

$$r_{ij}^t = \frac{\theta_j^t \phi(v_i; \mu_j^t, \Sigma_j^t)}{\sum_{l=1}^k \theta_l^t \phi(v_i; \mu_l^t, \Sigma_l^t)} \tag{3}$$

5:     **Maximization**
    Update the parameters of the model as:

$$\theta_j^{t+1} \leftarrow \frac{1}{n} \sum_{i=1}^n r_{ij}^t \tag{4}$$

$$\mu_j^{t+1} \leftarrow \frac{\sum_{i=1}^n r_{ij}^t v_i}{\sum_{i=1}^n r_{ij}^t} \tag{5}$$

$$\Sigma_j^{t+1} \leftarrow \frac{\sum_{i=1}^n r_{ij}^t (v_i - \mu_j^{t+1})(v_i - \mu_j^{t+1})^T}{\sum_{i=1}^n r_{ij}^t} \tag{6}$$

6:     t=t+1
7: **until**
8:

$$|\ell(\gamma^{t-1}; V) - \ell(\gamma^t; V)| < \tau \tag{7}$$

9: Return $\gamma^t = (\theta^t, \boldsymbol{\mu}^t, \boldsymbol{\Sigma}^t)$

---

*Proof.* We need to find the $K$ such that for all $x, y \in \mathbb{R}^d$, we have that $\|\sigma_j(y) - \sigma_j(x)\| \leq K \|y - x\|$. Observing that $\sigma_j$ is differentiable and that if we apply Cauchy-Schwarz to the statement of the Mean-Value-Theorem we derive that $\forall x, y \in U$, $\exists c$ such that $\|f(x) - f(y)\| \leq \|\bar{\nabla} f(c)\|_F \|x - y\|$. So to show Lipschitz continuity it is enough to select $K \leq \|\nabla \sigma_j\|_F^* = \max_{c \in \mathbb{R}^d} \|\nabla \sigma_j(c)\|$.

The partial derivatives $\frac{d\sigma_j(v)}{dv_i}$ are $\sigma_j(v)(1 - \sigma_j(v))$ if $i = j$ and $-\sigma_i(v)\sigma_j(v)$ otherwise. So $\|\nabla \sigma_j\|_F^2 = \sum_{i=1}^{d-1}(-\sigma(v)_i \sigma_j(v))^2 + \sigma_j(v)^2(1 - \sigma_j(v))^2 \leq \sum_{i=1}^{d-1} \sigma(v)_i \sigma_j(v) + \sigma_j(v)(1 - \sigma_j(v)) \leq \sigma_j(v) \sum_{i=0}^{d-1} \sigma_i(v) + 1 - \sigma_j(v) \leq 2\sigma_j(v) \leq 2$. In our case we can deduce that: $\|\sigma_j(y) - \sigma_j(x)\| \leq \sqrt{2} \|y - x\|$ so $K \leq \sqrt{2}$.

$\square$

**Claim 6.4.** *(Kerenidis & Prakash, 2017b) Let $\theta$ be the angle between vectors $x, y$, and assume that $\theta < \pi/2$. Then, $\|x - y\| \leq \epsilon$ implies $\| |x\rangle - |y\rangle \| \leq \frac{\sqrt{2}\epsilon}{\|x\|}$.*

We will also use Claim 4.5 from (Kerenidis et al., 2018).

**Claim 6.5.** *(Kerenidis et al., 2018) Let $\epsilon_b$ be the error we commit in estimating $|c\rangle$ such that $\| |c\rangle - |\bar{c}\rangle \| < \epsilon_b$, and $\epsilon_a$ the error we commit in the estimating the norms, $| \|c\| - \overline{\|c\|} | \leq \epsilon_a \|c\|$. Then $\|\bar{c} - c\| \leq \sqrt{\eta}(\epsilon_a + \epsilon_b)$.*

**Definition 2** (Exponential Family(Murphy, 2012))**.** *A probability density function or probability mass function $p(v|\nu)$ for $v = (v_1, \cdots, v_m) \in \mathcal{V}^m$, where $\mathcal{V} \subseteq \mathbb{R}$, $\nu \in \mathbb{R}^p$ is said to be in the exponential family if can be written as:*

$$p(v|\nu) := h(v) \exp\{o(\nu)^T T(v) - A(\nu)\}$$

*where:*

- $\nu \in \mathbb{R}^p$ *is called the* canonical or natural *parameter of the family,*

- $o(\nu)$ *is a function of* $\nu$ *(which often is just the identity function),*

- $T(v)$ *is the vector of sufficient statistics: a function that holds all the information the data* $v$ *holds with respect to the unknown parameters,*

- $A(\nu)$ *is the cumulant generating function, or log-partition function, which acts as a normalization factor,*

- $h(v) > 0$ *is the* base measure *which is a non-informative prior and de-facto is scaling constant.*

## 6.2 QUANTUM PROCEDURES

To prove our results, we are going to use the quantum procedures listed hereafter.

**Theorem 6.6** (Amplitude estimation and amplification (Brassard et al., 2002)). *If there is unitary operator* $U$ *such that* $U|0\rangle^l = |\phi\rangle = \sin(\theta)|x, 0\rangle + \cos(\theta)|G, 0^\perp\rangle$ *then* $\sin^2(\theta)$ *can be estimated to multiplicative error* $\eta$ *in time* $O(\frac{T(U)}{\eta \sin(\theta)})$ *and* $|x\rangle$ *can be generated in expected time* $O(\frac{T(U)}{\sin(\theta)})$.

We also need some state preparation procedures. These subroutines are needed for encoding vectors in $v_i \in \mathbb{R}^d$ into quantum states $|v_i\rangle$. An efficient state preparation procedure is provided by the QRAM data structures. We stress the fact that our result continues to hold, no matter how the efficient quantum loading of the data is provided. For instance, the data can be accessed through a QRAM, through a block encoding, or when the data can be produced by quantum circuits.

**Theorem 6.7** (QRAM data structure (Kerenidis & Prakash, 2017a)). *Let* $V \in \mathbb{R}^{n \times d}$*, there is a data structure to store the rows of* $V$ *such that,*

1. *The time to insert, update or delete a single entry* $v_{ij}$ *is* $O(\log^2(n))$.

2. *A quantum algorithm with access to the data structure can perform the following unitaries in time* $T = O(\log^2 N)$.

   (a) $|i\rangle|0\rangle \to |i\rangle|v_i\rangle$ *for* $i \in [n]$.
   (b) $|0\rangle \to \sum_{i \in [n]} \|v_i\| |i\rangle$.

In our algorithm we will also use subroutines for quantum linear algebra. For a symmetric matrix $M \in \mathbb{R}^{d \times d}$ with spectral norm $\|M\| = 1$ stored in the QRAM, the running time of these algorithms depends linearly on the condition number $\kappa(M)$ of the matrix, that can be replaced by $\kappa_\tau(M)$, a condition threshold where we keep only the singular values bigger than $\tau$, and the parameter $\mu(M)$, a matrix dependent parameter defined as

$$\mu(M) = \min_{p \in P}(\|M\|_F, \sqrt{s_{2p}(M)s_{2(1-p)}(M^T)}),$$

for $s_p(M) := \max_{i \in [n]} \|m_i\|_p^p$ where $\|m_i\|_p$ is the $\ell_p$ norm of the i-th row of $M$, and $P$ is a finte set of size $O(1) \in [0, 1]$. Note that $\mu(M) \le \|M\|_F \le \sqrt{d}$ as we have assumed that $\|M\| = 1$. The running time also depends logarithmically on the relative error $\epsilon$ of the final outcome state. (Chakraborty et al., 2018; Gilyén et al., 2018b).

**Theorem 6.8** (Quantum linear algebra (Chakraborty et al., 2018; Gilyén et al., 2018b) ). *Let* $M \in \mathbb{R}^{d \times d}$ *such that* $\|M\|_2 = 1$ *and* $x \in \mathbb{R}^d$*. Let* $\epsilon, \delta > 0$*. If* $M$ *is stored in appropriate QRAM data structures and the time to prepare* $|x\rangle$ *is* $T_x$*, then there exist quantum algorithms that with probability at least* $1 - 1/poly(d)$ *return a state* $|z\rangle$ *such that* $\||z\rangle - |Mx\rangle\| \le \epsilon$ *in time* $\widetilde{O}((\kappa(M)\mu(M) + T_x\kappa(M))\log(1/\epsilon))$.

**Theorem 6.9** (Quantum linear algebra for matrix products (Chakraborty et al., 2018) ). *Let* $M_1, M_2 \in \mathbb{R}^{d \times d}$ *such that* $\|M\|_1 = \|M\|_2 = 1$ *and* $x \in \mathbb{R}^d$*, and a vector* $x \in \mathbb{R}^d$ *stored in QRAM. Let* $\epsilon > 0$*. Then there exist quantum algorithms that with probability at least* $1 - 1/poly(d)$ *returns a state* $|z\rangle$ *such that* $\||z\rangle - |Mx\rangle\| \le \epsilon$ *in time* $\widetilde{O}((\kappa(M)(\mu(M_1)T_{M_1} + \mu(M_2)T_{M_2}))\log(1/\epsilon))$*, where* $T_{M_1}, T_{M_2}$ *is the time needed to index the rows of* $M_1$ *and* $M_2$.

The linear algebra procedures above can also be applied to any rectangular matrix $V \in \mathbb{R}^{N \times d}$ by considering instead the symmetric matrix $\overline{V} = \begin{pmatrix} 0 & V \\ V^T & 0 \end{pmatrix}$.

The final component needed for the $q$-means algorithm is a linear time algorithm for vector state tomography that will be used to recover classical information from the quantum states corresponding to the new centroids in each step. Given a unitary $U$ that produces a quantum state $|x\rangle$, by calling $O(d \log d/\epsilon^2)$ times $U$, the tomography algorithm is able to reconstruct a vector $\widetilde{x}$ that approximates $|x\rangle$ such that $\||\widetilde{x}\rangle - |x\rangle\| \leq \epsilon$.

**Theorem 6.10** (Vector state tomography (Kerenidis & Prakash, 2018)). *Given access to unitary $U$ such that $U|0\rangle = |x\rangle$ and its controlled version in time $T(U)$, there is a tomography algorithm with time complexity $O(T(U)\frac{d \log d}{\epsilon^2})$ that produces unit vector $\widetilde{x} \in \mathbb{R}^d$ such that $\|\widetilde{x} - x\|_2 \leq \epsilon$ with probability at least $(1 - 1/poly(d))$.*

**Lemma 6.11** (Distance / Inner Products Estimation (Kerenidis et al., 2018; Wiebe et al., 2014; Lloyd et al., 2013a)). *Assume for a data matrix $V \in \mathbb{R}^{N \times d}$ and a centroid matrix $C \in \mathbb{R}^{k \times d}$ that the following unitaries $|i\rangle |0\rangle \mapsto |i\rangle |v_i\rangle$, and $|j\rangle |0\rangle \mapsto |j\rangle |c_j\rangle$ can be performed in time $T$ and the norms of the vectors are known. For any $\Delta > 0$ and $\epsilon > 0$, there exists a quantum algorithm that computes*

$$|i\rangle |j\rangle |0\rangle \quad \mapsto \quad |i\rangle |j\rangle |\overline{d^2(v_i, c_j)}\rangle, \text{ where } |\overline{d^2(v_i, c_j)} - d^2(v_i, c_j)| \leqslant \epsilon \text{ with probability at least } 1 - 2\Delta, \text{ or}$$

$$|i\rangle |j\rangle |0\rangle \quad \mapsto \quad |i\rangle |j\rangle |\overline{(v_i, c_j)}\rangle, \text{ where } |\overline{(v_i, c_j)} - (v_i, c_j)| \leqslant \epsilon \text{ with probability at least } 1 - 2\Delta$$

*in time $\widetilde{O}\left(\frac{\|v_i\|\|c_j\|T \log(1/\Delta)}{\epsilon}\right)$.*

## 6.3 Initialization strategies for EM

Unlike k-means clustering, choosing a good set of initial parameters for a mixture of Gaussian is by no means trivial, and in multivariate context is known that the solution is problem-dependent. There are plenty of proposed techniques, and here we describe a few of them. Fortunately, these initialization strategies can be directly translated into quantum subroutines without impacting the overall running time of the quantum algorithm.

The simplest technique is called *random EM*, and consists in selecting initial points at random from the dataset as centroids, and sample the dataset to estimate the covariance matrix of the data. Then these estimates are used as the starting configuration of the model, and we may repeat the random sampling until we get satisfactory results.

A more standard technique borrows directly the initialization strategy of *k-means++* proposed in (Arthur & Vassilvitskii, 2007), and extends it to make an initial guess for the covariance matrices and the mixing weights. The initial guess for the centroids is selected by sampling from a suitable, easy to calculate distribution. This heuristic works as following: Let $c_0$ be a randomly selected point of the dataset, as first centroid. The other $k-1$ centroids are selected by selecting a vector $v_i$ with probability proportional to $d^2(v_i, \mu_{l(v_i)})$, where $\mu_{l(v_i)}$ is the previously selected centroid that is the closest to $v_i$ in $\ell_2$ distance. These centroids are then used as initial centroids for a round of k-means algorithm to obtain $\mu_1^0 \cdots \mu_j^0$. Then, the covariance matrices can be initialized as $\Sigma_j^0 := \frac{1}{|\mathcal{C}_j|} \sum_{i \in \mathcal{C}_j} (v_i - \mu_j)(v_i - \mu_j)^T$, where $\mathcal{C}_j$ is the set of samples in the training set that have been assigned to the cluster $j$ in the previous round of k-means. The mixing weights are estimated as $\mathcal{C}_j/n$. Eventually $\Sigma_j^0$ is regularized to be a PSD matrix.

There are other possible choices for parameter initialization in EM, for instance, based on *Hierarchical Agglomerative Clustering (HAC)* and the *CEM* algorithm. In CEM we run one step of EM, but with a so-called classification step between E and M. The classification step consists in a hard-clustering after computing the initial conditional probabilities (in the E step). The M step then calculates the initial guess of the parameters (Celeux & Govaert, 1992). In the *small EM* initialization method we run EM with a different choice of initial parameters using some of the previous strategies. The difference here is that we repeat the EM algorithm for a few numbers of iterations, and we keep iterating from the choice of parameters that returned the best partial results. For an overview and comparison of different initialization techniques, we refer to (Blömer & Bujna, 2013; Biernacki et al., 2003).

**Quantum initialization strategies** For the initialization of $\gamma^0$ in the quantum algorithm we can use the same initialization strategies as in classical machine learning. For instance, we can use the classical *random EM* initialization strategy for QEM.

A quantum initialization strategy can also be given using the *k-means++* initializion strategy from (Kerenidis et al., 2018), which returns $k$ initial guesses for the centroids $c_1^0 \cdots c_k^0$ consistent with the classical algorithm in time $\left(k^2 \frac{2\eta^{1.5}}{\epsilon \sqrt{\mathbb{E}(d^2(v_i, v_j))}}\right)$, where $\mathbb{E}(d^2(v_i, v_j))$ is the average squared distance between two points of the dataset, and $\epsilon$ is the tolerance in the distance estimation. From there, we can perform a full round of q-means algorithm and get an

estimate for $\mu_1^0 \cdots \mu_k^0$. With q-means and the new centroids store in the QRAM we can create the state

$$|\psi^0\rangle := \frac{1}{\sqrt{n}} \sum_{i=1}^{n} |i\rangle \, |l(v_i)\rangle \,. \tag{8}$$

Where $l(v_i)$ is the label of the closest centroid to the $i$-th point. By sampling $S \in O(d)$ points from this state we get two things. First, from the frequency $f_j$ of the second register we can have an guess of $\theta_j^0 \leftarrow |\mathcal{C}_j|/n \sim f_j/S$. Then, from the first register we can estimate $\Sigma_j^0 \leftarrow \sum_{i \in S} (v_i - \mu_j^0)(v_i - \mu_j^0)^T$. Sampling $O(d)$ points and creating the state in Equation (8) takes time $\widetilde{O}(dk\eta)$ by Theorem 6.11 and the minimum finding procedure described in (Kerenidis et al., 2018).

Techniques illustrated in (Miyahara et al., 2019) can also be used to quantize the CEM algorithm which needs a hard-clustering step. Among the different possible approaches, the *random* and the *small EM* greatly benefit from a faster algorithm, as we can spend more time exploring the space of the parameters by starting from different initial seeds, and thus avoid local minima of the likelihood.

## 6.4 Special cases of GMM.

What we presented in the previous section is the most general model of GMM. For simple datasets, it is common to assume some restrictions on the covariance matrices of the mixtures. The translation into a quantum version of the model should be straightforward. We distinguish between these cases:

1. **Soft $k$-means**. This algorithm is often presented as a generalization of k-means, but it can actually be seen as special case of EM for GMM - albeit with a different assignment rule. In soft $k$-means, the assignment function is replaced by a softmax function with *stiffness* parameter $\beta$. This $\beta$ represents the covariance of the clusters. It is assumed to be equal for all the clusters, and for all dimensions of the feature space. Gaussian Mixtures with constant covariance matrix (i.e. $\Sigma_j = \beta I$ for $\beta \in \mathbb{R}$) can be interpreted as a kind of soft or fuzzy version of k-means clustering. The probability of a point in the feature space being assigned to a certain cluster $j$ is:

$$r_{ij} = \frac{e^{-\beta \|x_i - \mu_i\|^2}}{\sum_{l=1}^{k} e^{-\beta \|x_i - \mu_l\|^2}}$$

   where $\beta > 0$ is the stiffness parameter.

2. **Spherical**. In this model, each component has its own covariance matrix, but the variance is uniform in all the directions, thus reducing the covariance matrix to a multiple of the identity matrix (i.e. $\Sigma_j = \sigma_j^2 I$ for $\sigma_j \in \mathbb{R}$).

3. **Diagonal**. As the name suggests, in this special case the covariance matrix of the distributions is a diagonal matrix, but different Gaussians might have different diagonal covariance matrices.

4. **Tied**. In this model, the Gaussians share the same covariance matrix, without having further restriction on the Gaussian.

5. **Full**. This is the most general case, where each of the components of the mixture have a different, SDP, covariance matrix.

## 6.5 Proofs

**Lemma 6.12** (Determinant evaluation). *There is an algorithm that, given as input a matrix $\Sigma$ and a parameter $0 < \delta < 1$, outputs an estimate $\overline{\log(det(\Sigma))}$ such that $|\overline{\log(det(\Sigma))} - \log(det(\Sigma))| \leq \epsilon$ with probability $1 - \delta$ in time:*

$$T_{Det,\epsilon} = \widetilde{O}\left(\epsilon^{-2}\kappa(\Sigma)\log(1/\delta)nnz(\Sigma)|\log(det(\Sigma))|\right)$$

*Proof.* In order to apply Theorem 6.1, we need to be sure that all the eigenvalues lie in $(\sigma_{min}, 1)$. In order to satisfy this condition, we can scale the matrix by a constant factor $c$, such that $\Sigma' = \Sigma/c$. In this way, $\log det(\Sigma') = \log \prod_i^d (\sigma_i/c)$. Therefore, $\log(det(\Sigma')) = \log(det(\Sigma)) - \log(c^d)$. This will allow us to recover the value of $\log det(\Sigma)$ by using

Theorem 6.1. We apply the Theorem with precision $\epsilon = 1/4$ to get an estimate $\gamma$ such that $\frac{1}{2} \le \frac{\gamma}{\log(\det(\Sigma))} \le \frac{3}{2}$. Then, to have an estimate with absolute error $\epsilon$, we apply Theorem 6.1 with precision $\epsilon' = \frac{\epsilon}{4\gamma}$. This gives us an estimate for $\log(\det(\Sigma))$ with error $2\epsilon' \log(\det(\Sigma)) \le \epsilon$ in time:

$$\widetilde{O}\left(\epsilon^{-2}\kappa(\Sigma)\log(1/\delta)nnz(\Sigma)|\log(\det(\Sigma))|\right).$$

$\square$

**Lemma 6.13** (Error in the responsibilities of the exponential family). *Let $v_i \in \mathbb{R}^n$ be a vector, and let $\{p(v_i|\nu_j)\}_{j=1}^k$ be a set of $k$ probability distributions in the exponential family, defined as $p(v_i|\nu_j) := h_j(v_i)exp\{o_j(\nu_j)^T T_j(v_i) - A_j(\nu_j)\}$. Then, if we have estimates for each exponent with error $\epsilon$, then we can compute each $r_{ij}$ such that $|\overline{r_{ij}} - r_{ij}| \le \sqrt{2k}\epsilon$ for $j \in [k]$.*

*Proof.* The proof follows from rewriting the responsibility of Equation (3) as:

$$r_{ij} := \frac{h_j(v_i)\exp\{o_j(\nu_j)^T T(v_i) - A_j(\nu_j) + \log\theta_j\}}{\displaystyle\sum_{l=1}^{k} h_l(v_i)\exp\{o_l(\nu_l)^T T(v_i) - A_l(\nu_l) + \log\theta_l\}} \tag{9}$$

In this form, it is clear that the responsibilities can be seen a *softmax* function, and we can use Theorem 6.3 to bound the error in computing this value.

Let $T_i \in \mathbb{R}^k$ be the vector of the exponent, that is $t_{ij} = o_j(\nu_j)^T T(v_i) - A_j(\nu_j) + \log\theta_j$. In an analogous way we define $\overline{T_i}$ the vector where each component is the estimate with error $\epsilon$. The error in the responsibility is defined as $|r_{ij} - \overline{r_{ij}}| = |\sigma_j(T_i) - \sigma_j(\overline{T_i})|$. Because the function $\sigma_j$ is Lipschitz continuous, as we proved in Theorem 6.3 with a Lipschitz constant $K \le \sqrt{2}$, we have that, $|\sigma_j(T_i) - \sigma_j(\overline{T_i})| \le \sqrt{2}\left\|T_i - \overline{T_i}\right\|$. The result follows as $\left\|T_i - \overline{T_i}\right\| < \sqrt{k}\epsilon$. $\square$

**Lemma 6.14** (Quantum Gaussian Evaluation). *Suppose we have stored in the QRAM a matrix $V \in \mathbb{R}^{n \times d}$, the centroid $\mu \in \mathbb{R}^d$ and the covariance matrix $\Sigma \in \mathbb{R}^{d \times d}$ of a multivariate Gaussian distribution $\phi(v|\mu, \Sigma)$, as well as an estimate for $\log(\det(\Sigma))$. Then for $\epsilon_1 > 0$, there exists a quantum algorithm that with probability $1 - \gamma$ performs the mapping,*

- $U_{G,\epsilon_1} : |i\rangle|0\rangle \to |i\rangle|\overline{s_i}\rangle$ *such that $|s_i - \overline{s_i}| < \epsilon_1$, where $s_i = -\frac{1}{2}((v_i - \mu)^T \Sigma^{-1}(v_i - \mu) + d\log 2\pi + \log(det(\Sigma)))$ is the exponent for the Gaussian probability density function.*

*The running time of the algorithm is,*

$$T_{G,\epsilon_1} = O\left(\frac{\kappa^2(\Sigma)\mu(\Sigma)\log(1/\gamma)}{\epsilon_1}\eta\right). \tag{10}$$

*Proof.* We use quantum linear algebra and inner product estimation to estimate the quadratic form $(v_i - \mu)^T \Sigma^{-1}(v_i - \mu)$ to error $\epsilon_1$. First, we decompose the quadratic form as $v_i^T \Sigma^{-1} v_i - 2v_i^T \Sigma^{-1}\mu + \mu^T \Sigma^{-1}\mu$ and separately approximate each term in the sum to error $\epsilon_1/4$.

We describe the procedure to estimate $\mu^T \Sigma^{-1} v_i = \left\|\Sigma^{-1} v_i\right\| \|\mu\| \langle\mu|\Sigma^{-1} v_i\rangle$, the other estimates are obtained similarly. We use the quantum linear algebra subroutines in Theorem 6.8 to construct $|\Sigma^{-1} v_i\rangle$ up to error $\epsilon_3 \ll \epsilon_1$ in time $O(\kappa(\Sigma)\mu(\Sigma)\log(1/\epsilon_3))$ and estimate $\left\|\Sigma^{-1} v_i\right\|$ up to error $\epsilon_1$ in time $O(\kappa(\Sigma)\mu(\Sigma)\log(1/\epsilon_3)/\epsilon_1)$ which gives us the mapping $|i\rangle|0\rangle \mapsto |i\rangle|\overline{\|\Sigma^{-1} v_i\|}\rangle$. We then use quantum inner product estimation (Theorem 6.11) to estimate $\langle\mu, \Sigma^{-1} v_i\rangle$ to additive error $\frac{\epsilon_1}{4\|\mu\|\|\Sigma^{-1} v_i\|}$. The procedure estimates $(\mu, \Sigma^{-1} v_i)$ within additive error $\epsilon_1/4$. The procedure succeeds with probability $1 - \gamma$ and requires time $O(\frac{\kappa(\Sigma)\mu(\Sigma)\log(1/\gamma)\log(1/\epsilon_3)}{\epsilon_1}\|\mu\|\left\|\Sigma^{-1} v_i\right\|)$. Using similar estimation procedure for $v_i^T \Sigma^{-1}\mu$ and $\mu^T \Sigma^{-1}\mu$, we obtain an estimate for $\frac{1}{2}((v_i - \mu)^T \Sigma^{-1}(v_i - \mu)$ within error $\epsilon_1$.

Recall that (through Lemma 3.1) we also have an estimate of the log-determinant to error $\epsilon_1$. Thus we obtain an approximation for $-\frac{1}{2}((v_i - \mu)^T \Sigma^{-1}(v_i - \mu) + d\log 2\pi + \log(\det(\Sigma)))$ within error $2\epsilon_1$. We have the upper bound,

$\left\| \Sigma^{-1} v_i \right\| \leq \left\| \Sigma^{-1} \right\| \left\| v_i \right\| \leq \kappa(\Sigma) \left\| v_i \right\|$, as $\|\Sigma\| = 1$ for $\Sigma$ stored in a QRAM data structure. Further observing that $\|u\| \leq \sqrt{\eta}$ and $\|v_i\| \leq \sqrt{\eta}$, the running time for this computation is $O\left( \frac{\kappa^2(\Sigma)\mu(\Sigma)\log(1/\gamma)\log(1/\epsilon_3)}{\epsilon_1} \eta \right)$.

$\square$

**Lemma 6.15** (Calculating responsibilities). *Suppose we have quantum access to a GMM with parameters $\gamma^t = (\theta^t, \boldsymbol{\mu}^t, \boldsymbol{\Sigma}^t)$. There are quantum algorithms that can:*

1. *Perform the mapping $|i\rangle |j\rangle |0\rangle \mapsto |i\rangle |j\rangle |\overline{r_{ij}}\rangle$ such that $|\overline{r_{ij}} - r_{ij}| \leq \epsilon_1$ with probability $1 - \gamma$ in time:*
$$T_{R_1,\epsilon_1} = \widetilde{O}(k^{1.5} \times T_{G,\epsilon_1})$$

2. *For a given $j \in [k]$, construct state $|\overline{R_j}\rangle$ such that $\left\| |\overline{R_j}\rangle - \frac{1}{\sqrt{Z_j}} \sum_{i=0}^{n} r_{ij} |i\rangle \right\| < \epsilon_1$ where $Z_j = \sum_{i=0}^{n} r_{ij}^2$ with high probability in time:*
$$T_{R_2,\epsilon_1} = \widetilde{O}(k^2 \times T_{R_1,\epsilon_1})$$

*Proof.* For the first statement, let's recall the definition of responsibility: $r_{ij} = \frac{\theta_j \phi(v_i;\mu_j,\Sigma_j)}{\sum_{l=1}^{k} \theta_l \phi(v_i;\mu_l,\Sigma_l)}$. With the aid of $U_{G,\epsilon_1}$ of Lemma 3.2 we can estimate $\log(\phi(v_i|\mu_j,\Sigma_j))$ for all $j$ up to additive error $\epsilon_1$, and then using the current estimate of $\theta^t$, we can calculate the responsibilities create the state,

$$\frac{1}{\sqrt{n}} \sum_{i=0}^{n} |i\rangle \left( \bigotimes_{j=1}^{k} |j\rangle |\overline{\log(\phi(v_i|\mu_j,\Sigma_j))}\rangle \right) \otimes |\overline{r_{ij}}\rangle .$$

The estimate $\overline{r_{ij}}$ is computed by evaluating a weighted softmax function with arguments $\overline{\log(\phi(v_i|\mu_j,\Sigma_j))}$ for $j \in [k]$. The estimates $\overline{\log(\phi(v_i|\mu_j,\Sigma_j))}$ are then uncomputed. The runtime of the procedure is given by calling $k$ times Lemma 3.2 for Gaussian estimation (the arithmetic operations to calculate the responsibilities are absorbed).

Let us analyze the error in the estimation of $r_{ij}$. The responsibility $r_{ij}$ is a softmax function with arguments $\log(\phi(v_i|\mu_j,\Sigma_j))$ that are computed upto error $\epsilon_1$ using Lemma 3.2. As the softmax function has a Lipschitz constant $K \leq \sqrt{2}$ by Lemma 6.13, we choose precision for Lemma 3.2 to be $\epsilon_1/\sqrt{2k}$ to get the guarantee $|\overline{r_{ij}} - r_{ij}| \leq \epsilon_1$. Thus, the total cost of this step is $T_{R_1,\epsilon_1} = k^{1.5} T_{G,\epsilon_1}$.

Now let's see how to encode this information in the amplitudes, as stated in the second claim of the Lemma. We estimate the responsibilities $r_{ij}$ to some precision $\epsilon$ and perform a controlled rotation on an ancillary qubit to obtain,

$$\frac{1}{\sqrt{n}} |j\rangle \sum_{i=0}^{n} |i\rangle |\overline{r_{ij}}\rangle \left( \overline{r_{ij}} |0\rangle + \sqrt{1 - \overline{r_{ij}}^2} |1\rangle \right). \tag{11}$$

We then undo the circuit on the second register and perform amplitude amplification on the rightmost auxiliary qubit being $|0\rangle$ to get $|\overline{R_j}\rangle := \frac{1}{\|\overline{R_j}\|} \sum_{i=0}^{n} \overline{r_{ij}} |i\rangle$. The runtime for amplitude amplification on this task is $O(T_{R_1,\epsilon} \cdot \frac{\sqrt{n}}{\|\overline{R_j}\|})$.

Let us analyze the precision $\epsilon$ required to prepare $|\overline{R_j}\rangle$ such that $\left\| |R_j\rangle - |\overline{R_j}\rangle \right\| \leq \epsilon_1$. As we have estimates $|r_{ij} - \overline{r_{ij}}| < \epsilon$ for all $i,j$, the $\ell_2$-norm error $\|R_j - \overline{R_j}\| = \sqrt{\sum_{i=0}^{n} |r_{ij} - \overline{r_{ij}}|^2} < \sqrt{n}\epsilon$. Applying Claim 6.4, the error for the normalized vector $|R_j\rangle$ can be bounded as $\left\| |R_j\rangle - |\overline{R_j}\rangle \right\| < \frac{\sqrt{2n}\epsilon}{\|R_j\|}$. By the Cauchy-Schwarz inequality we have that $\|R_j\| \geq \frac{\sum_i^n r_{ij}}{\sqrt{n}}$. We can use this to obtain a bound $\frac{\sqrt{n}}{\|R_j\|} < \frac{\sqrt{n}}{\sum_i r_{ij}}\sqrt{n} = O(k)$, using the dataset assumptions in section 2. If we choose $\epsilon$ such that $\frac{\sqrt{2n}\epsilon}{\|R_j\|} < \epsilon_1$, that is $\epsilon \leq \epsilon_1/k$ then our runtime becomes $T_{R_2,\epsilon_1} := \widetilde{O}(k^2 \times T_{R_1,\epsilon_1})$.

$\square$

**Lemma 6.16** (Computing $\theta^{t+1}$). *We assume quantum access to a GMM with parameters $\gamma^t$ and let $\delta_\theta > 0$ be a precision parameter. There exists an algorithm that estimates $\overline{\theta}^{t+1} \in \mathbb{R}^k$ such that $\left\| \overline{\theta}^{t+1} - \theta^{t+1} \right\| \leq \delta_\theta$ in time*

$$T_\theta = O\left( k^{3.5}\eta^{1.5} \frac{\kappa^2(\Sigma)\mu(\Sigma)}{\delta_\theta^2} \right)$$

*Proof.* An estimate of $\theta_j^{t+1}$ can be recovered from the following operations. First, we use Lemma 3.3 (part 1) to compute the responsibilities to error $\epsilon_1$, and then perform the following mapping, which consists of a controlled rotation on an auxiliary qubit:

$$\frac{1}{\sqrt{nk}} \sum_{\substack{i=1 \\ j=1}}^{n,k} |i\rangle |j\rangle |\overline{r_{ij}}^t\rangle \mapsto \frac{1}{\sqrt{nk}} \sum_{\substack{i=1 \\ j=1}}^{n,k} |i\rangle |j\rangle \left( \sqrt{\overline{r_{ij}}^t} |0\rangle + \sqrt{1 - \overline{r_{ij}}^t} |1\rangle \right)$$

The previous operation has a cost of $T_{R_1,\epsilon_1}$, and the probability of getting $|0\rangle$ is $p(0) = \frac{1}{nk} \sum_{i=1}^n \sum_{j=1}^k r_{ij}^t = \frac{1}{k}$.

Recall that $\theta_j^{t+1} = \frac{1}{n} \sum_{i=1}^n r_{ij}^t$ by definition. Let $Z_j = \sum_{i=1}^n \overline{r_{ij}}^t$ and define state $|\sqrt{R_j}\rangle = \left( \frac{1}{\sqrt{Z_j}} \sum_{i=1}^n \sqrt{\overline{r_{ij}}^t} |i\rangle \right) |j\rangle$. After amplitude amplification on $|0\rangle$ we have the state,

$$|\sqrt{R}\rangle := \frac{1}{\sqrt{n}} \sum_{\substack{i=1 \\ j=1}}^{n,k} \sqrt{\overline{r_{ij}}^t} |i\rangle |j\rangle$$

$$= \sum_{j=1}^k \sqrt{\frac{Z_j}{n}} \left( \frac{1}{\sqrt{Z_j}} \sum_{i=1}^n \sqrt{\overline{r_{ij}}^t} |i\rangle \right) |j\rangle$$

$$= \sum_{j=1}^k \sqrt{\overline{\theta_j}^{t+1}} |\sqrt{R_j}\rangle . \qquad (12)$$

The probability of obtaining outcome $|j\rangle$ if the second register is measured in the standard basis is $p(j) = \overline{\theta_j}^{t+1}$.

An estimate for $\theta_j^{t+1}$ with precision $\epsilon$ can be obtained by either sampling the last register, or by performing amplitude estimation to estimate each of the values $\theta_j^{t+1}$ for $j \in [k]$. Sampling requires $O(\epsilon^{-2})$ samples by the Chernoff bounds, but does not incur any dependence on $k$. In this case, as the number of cluster $k$ is relatively small compared to $1/\epsilon$, we chose to do amplitude estimation to estimate all $\theta_j^{t+1}$ for $j \in [k]$ to error $\epsilon/\sqrt{k}$ in time,

$$T_\theta := O\left( k \cdot \frac{\sqrt{k} T_{R_1,\epsilon_1}}{\epsilon} \right) . \qquad (13)$$

Let's analyze the error in the estimation of $\theta_j^{t+1}$. For the error due to responsibility estimation by Lemma 3.3 we have $|\overline{\theta_j}^{t+1} - \theta_j^{t+1}| = \frac{1}{n} \sum_i |\overline{r_{ij}}^t - r_{ij}^t| \leq \epsilon_1$ for all $j \in [k]$, implying that $\left\| \overline{\theta}^{t+1} - \theta^{t+1} \right\| \leq \sqrt{k}\epsilon_1$. The total error in $\ell_2$ norm due to Amplitude estimation is at most $\epsilon$ as it estimates each coordinate of $\overline{\theta_j}^{t+1}$ to error $\epsilon/\sqrt{k}$.

Using the triangle inequality, we have the total error is at most $\epsilon + \sqrt{k}\epsilon_1$. As we require that the final error be upper bounded as $\left\| \overline{\theta}^{t+1} - \theta^{t+1} \right\| < \delta_\theta$, we choose parameters $\sqrt{k}\epsilon_1 < \delta_\theta/2 \Rightarrow \epsilon_1 < \frac{\delta_\theta}{2\sqrt{k}}$ and $\epsilon < \delta_\theta/2$. With these parameters, the overall running time of the quantum procedure is $T_\theta = O(k^{1.5} \frac{T_{R_1,\epsilon_1}}{\epsilon}) = O\left( k^{3.5} \frac{\eta^{1.5} \cdot \kappa^2(\Sigma)\mu(\Sigma)}{\delta_\theta^2} \right)$.

$\square$

**Lemma 6.17** (Computing $\boldsymbol{\mu}_j^{t+1}$). *We assume we have quantum access to a GMM with parameters $\gamma^t$. For a precision parameter $\delta_\mu > 0$, there is a quantum algorithm that calculates $\{\overline{\mu_j}^{t+1}\}_{j=1}^k$ such that for all $j \in [k]$ $\left\| \overline{\mu_j}^{t+1} - \mu_j^{t+1} \right\| \leq \delta_\mu$ in time*

$$T_\mu = \widetilde{O}\left( \frac{kd\eta\kappa(V)(\mu(V) + k^{3.5}\eta^{1.5}\kappa^2(\Sigma)\mu(\Sigma))}{\delta_\mu^3} \right)$$

*Proof.* The new centroid $\mu_j^{t+1}$ is estimated by first creating an approximation of the state $|R_j^t\rangle$ up to error $\epsilon_1$ in the $\ell_2$-norm using part 2 of Lemma 3.3. We then use the quantum linear algebra algorithms in Theorem 6.8 to multiply $R_j$ by

$V^T$, and compute a state $|\overline{\mu_j}^{t+1}\rangle$ along with an estimate for the norm $\left\|V^T R_j^t\right\| = \left\|\overline{\mu_j}^{t+1}\right\|$ with error $\epsilon_{norm}$. The last step of the algorithm consists in estimating the unit vector $|\overline{\mu_j}^{t+1}\rangle$ with precision $\epsilon_{tom}$ using tomography. Considering that the tomography depends on $d$, which we expect to be bigger than the precision required by the norm estimation, we can assume that the runtime of the norm estimation is absorbed. Thus, we obtain: $\widetilde{O}\left(k\frac{d}{\epsilon_{tom}^2} \cdot \kappa(V)\left(\mu(V) + T_{R_2,\epsilon_1}\right)\right)$.

Let's now analyze the total error in the estimation of the new centroids, which we want to be $\delta_\mu$. For this purpose, we use Claim 6.5, and choose parameters such that $2\sqrt{\eta}(\epsilon_{tom} + \epsilon_{norm}) = \delta_\mu$. Since the error $\epsilon_3$ for quantum linear algebra appears as a logarithmic factor in the running time, we can choose $\epsilon_3 \ll \epsilon_{tom}$ without affecting the runtime.

Let $\overline{\mu}$ be the classical unit vector obtained after quantum tomography, and $\widehat{|\mu\rangle}$ be the state produced by the quantum linear algebra procedure starting with an approximation of $|R_j^t\rangle$. Using the triangle inequality we have $\||\mu\rangle - \overline{\mu}\| < \left\|\overline{\mu} - \widehat{|\mu\rangle}\right\| + \left\|\widehat{|\mu\rangle} - |\mu\rangle\right\| < \epsilon_{tom} + \epsilon_1 < \delta_\mu/2\sqrt{\eta}$. The errors for the norm estimation procedure can be bounded similarly as $| \|\mu\| - \overline{\|\mu\|}| < | \|\mu\| - \widehat{\|\mu\|}| + |\widehat{\|\mu\|} - \overline{\|\mu\|}| < \epsilon_{norm} + \epsilon_1 \le \delta_\mu/2\sqrt{\eta}$. We therefore choose parameters $\epsilon_{tom} = \epsilon_1 = \epsilon_{norm} \le \delta_\mu/4\sqrt{\eta}$. Since the amplitude estimation step we use for estimating the norms does not depends on $d$, which is expected to dominate the other parameters, we omit the amplitude estimation step. Substituting for $T_{R_2,\delta_\mu}$, we have the more concise expression for the running time of:

$$\widetilde{O}\left(\frac{kd\eta\kappa(V)(\mu(V) + k^{3.5}\eta^{1.5}\kappa^2(\Sigma)\mu(\Sigma))}{\delta_\mu^3}\right) \tag{14}$$

$\square$

**Lemma 6.18** (Computing $\Sigma_j^{t+1}$). *We assume we have quantum access to a GMM with parameters $\gamma^t$. We also have computed estimates $\overline{\mu_j}^{t+1}$ of all centroids such that $\left\|\overline{\mu_j}^{t+1} - \mu_j^{t+1}\right\| \le \delta_\mu$ for precision parameter $\delta_\mu > 0$. Then, there exists a quantum algorithm that outputs estimates for the new covariance matrices $\{\overline{\Sigma}_j^{t+1}\}_{j=1}^k$ such that $\left\|\Sigma_j^{t+1} - \overline{\Sigma}_j^{t+1}\right\|_F \le \delta_\mu\sqrt{\eta}$ with high probability, in time,*

$$T_\Sigma := \widetilde{O}\left(\frac{kd^2\eta\kappa^2(V)(\mu(V') + \eta^2 k^{3.5}\kappa^2(\Sigma)\mu(\Sigma))}{\delta_\mu^3}\right)$$

*Proof.* It is simple to check, that the update rule of the covariance matrix during the maximization step can be reduced to (Murphy, 2012, Exercise 11.2):

$$\Sigma_j^{t+1} \leftarrow \frac{\sum_{i=1}^n r_{ij}(v_i - \mu_j^{t+1})(v_i - \mu_j^{t+1})^T}{\sum_{i=1}^n r_{ij}} = \frac{\sum_{i=1}^n r_{ij}v_iv_i^T}{n\theta_j} - \mu_j^{t+1}(\mu_j^{t+1})^T = \Sigma_j' - \mu_j^{t+1}(\mu_j^{t+1})^T \tag{15}$$

First, note that we can use the estimates of the centroids to compute $\mu_j^{t+1}(\mu_j^{t+1})^T$ with error $\delta_\mu \|\mu\| \le \delta_\mu\sqrt{\eta}$ in the update rule for the $\Sigma_j$. This follows from the fact that $\overline{\mu} = \mu + e$ where $e$ is a vector of norm $\delta_\mu$. Therefore $\|\mu\mu^T - \overline{\mu}\,\overline{\mu}^T\| < 2\sqrt{\eta}\delta_\mu + \delta_\mu^2 \le 3\sqrt{\eta}\delta_\mu$. It follows that we can allow an error of $\sqrt{\eta}\delta_\mu$ also for the left term in the definition of $\Sigma_j^{t+1}$. Let's discuss the procedure for estimating $\Sigma_j'$ in Eq. (15). Note that $\text{vec}[\Sigma_j'] = (V')^T R_j$, so we use quantum matrix multiplication to estimate $|\text{vec}[\Sigma_j']\rangle$ and $\|\text{vec}[\Sigma_j']\|$. As the runtime for the norm estimation $\frac{\kappa(V')(\mu(V') + T_{R_2,\epsilon_1}))\log(1/\epsilon_{mult})}{\epsilon_{norms}}$ does not depend on $d$, we consider it smaller than the runtime for performing tomography. Thus, the runtime for this operation is:

$$O(\frac{d^2\log d}{\epsilon_{tom}^2}\kappa(V')(\mu(V') + T_{R_2,\epsilon_1}))\log(1/\epsilon_{mult})).$$

Let's analyze the error of this procedure. We want a matrix $\overline{\Sigma_j'}$ that is $\sqrt{\eta}\delta_\mu$-close to the correct one: $\left\|\overline{\Sigma_j'} - \Sigma_j'\right\|_F = \left\|\text{vec}[\overline{\Sigma_j'}] - \text{vec}[\Sigma_j']\right\|_2 < \sqrt{\eta}\delta_\mu$. Again, the error due to matrix multiplication can be taken as small as necessary,

since is inside a logarithm. From Claim 6.5, we just need to fix the error of tomography and norm estimation such that $\eta(\epsilon_{unit} + \epsilon_{norms}) < \sqrt{\eta}\delta_\mu$ where we have used $\eta$ as an upper bound on $\|\Sigma_j\|_F$. For the unit vectors, we require $\left\| |\Sigma_j'\rangle - \overline{|\Sigma_j'\rangle} \right\| \leq \left\| \overline{|\Sigma_j'\rangle} - \widehat{|\Sigma_j'\rangle} \right\| + \left\| \widehat{|\Sigma_j'\rangle} - |\Sigma_j'\rangle \right\| < \epsilon_{tom} + \epsilon_1 \leq \delta_\mu/2\sqrt{\eta}$, where $\overline{|\Sigma_j'\rangle}$ is the error due to tomography and $\widehat{|\Sigma_j'\rangle}$ is the error due to Lemma 3.3. For this inequality to be true, we choose $\epsilon_{tom} = \epsilon_1 < \delta_\mu/4\sqrt{\eta}$.

The same argument applies to estimating the norm $\|\Sigma_j'\|$ with relative error : $\big| \|\Sigma_j'\| - \overline{\|\Sigma_j'\|} \big| \leq |\overline{\|\Sigma_j'\|} - \widehat{\|\Sigma_j'\|}| + |\widehat{\|\Sigma_j'\|} - \|\Sigma_j'\| | < \epsilon + \epsilon_1 \leq \delta_\mu/2\sqrt{\eta}$ (where here $\epsilon$ is the error of the amplitude estimation step used in Theorem 6.8 and $\epsilon_1$ is the error in calling Lemma 3.3. Again, we choose $\epsilon = \epsilon_1 \leq \delta_\mu/4\sqrt{\eta}$. Note that $\kappa(V') \leq \kappa^2(V)$. This can be derived from the fact that $\kappa(A \otimes B) = \kappa(A)\kappa(B)$, $\kappa(AB) \leq \kappa(A)\kappa(B)$, and

$$V' := \begin{pmatrix} [e_1 \otimes e_1]^T \\ \vdots \\ [e_n \otimes e_n]^T \end{pmatrix} (V \otimes V).$$

Since the tomography is more costly than the amplitude estimation step, we can disregard the runtime for the norm estimation step. As this operation is repeated $k$ times for the $k$ different covariance matrices, the total runtime of the whole algorithm is given by $\widetilde{O}\left( \frac{kd^2\eta\kappa^2(V)(\mu(V')+\eta^2 k^{3.5}\kappa^2(\Sigma)\mu(\Sigma))}{\delta_\mu^3} \right)$.

Let us also note that for each of new computed covariance matrices, we use Lemma 3.1 to compute an estimate for their log-determinant and this time can be absorbed in the time $T_\Sigma$.

$\square$

**Lemma 6.19** (Quantum estimation of likelihood). *We assume we have quantum access to a GMM with parameters $\gamma^t$. For $\epsilon_\tau > 0$, there exists a quantum algorithm that estimates $\mathbb{E}[p(v_i; \gamma^t)]$ with absolute error $\epsilon_\tau$ in time*

$$T_\ell = \widetilde{O}\left( k^{1.5}\eta^{1.5} \frac{\kappa^2(\Sigma)\mu(\Sigma)}{\epsilon_\tau^2} \right)$$

*Proof.* We obtain the likelihood from the ability to compute the value of a Gaussian distribution and quantum arithmetic. Using the mapping of Lemma 3.2 with precision $\epsilon_1$, we can compute $\phi(v_i|\mu_j, \Sigma_j)$ for all the Gaussians, that is $|i\rangle \bigotimes_{j=0}^{k-1} |j\rangle \overline{|p(v_i|j; \gamma_j)\rangle}$. Then, by knowing $\theta$, and by using quantum arithmetic we can compute in a register the mixture of Gaussian's: $p(v_i; \gamma) = \sum_{j\in[k]} \theta_j p(v_i|j; \gamma)$. We now drop the notation for the model $\gamma$ and write $p(v_i)$ instead of $p(v_i; \gamma)$. Doing the previous calculations quantumly leads to the creation of the state $|i\rangle |p(v_i)\rangle$. We perform the mapping $|i\rangle |p(v_i)\rangle \mapsto |i\rangle \left( \sqrt{p(v_i)} |0\rangle + \sqrt{1 - p(v_i)} |1\rangle \right)$ and estimate $p(|0\rangle) \simeq \mathbb{E}[p(v_i)]$ with amplitude estimation on the ancilla qubit. To get a $\epsilon_\tau$-estimate of $p(0)$ we need to decide the precision parameter we use for estimating $\overline{p(v_i|j; \gamma)}$ and the precision required by amplitude estimation. Let $\overline{p(0)}$ be the $\epsilon_1$-error introduced by using Lemma 3.2 and $\widehat{p(0)}$ the error introduced by amplitude estimation. Using triangle inequality we set $\left\| p(0) - \widehat{p(0)} \right\| < \left\| \widehat{p(0)} - \overline{p(0)} \right\| + \left\| \overline{p(0)} - p(0) \right\| < \epsilon_\tau$.

To have $|p(0) - \overline{p(0)}| < \epsilon_\tau$, we should set $\epsilon_1$ such that $|\overline{p(0)} - p(0)| < \epsilon_\tau/4$, and we set the error in amplitude estimation and in the estimation of the probabilities to be $\epsilon_\tau/2$. The runtime of this procedure is therefore:

$$\widetilde{O}\left( k \cdot T_{G,\epsilon_\tau} \cdot \frac{1}{\epsilon_\tau \sqrt{p(0)}} \right) = \widetilde{O}\left( k^{1.5}\eta^{1.5} \cdot \frac{\kappa(\Sigma)\mu(\Sigma)}{\epsilon_\tau^2} \right)$$

$\square$

## 6.6 EXPERIMENTS

We used a subset of the voices that can be found on VoxForge (Voxforge.org). The training set consist in 5 speech utterances for 34 speakers (i.e. clips of a few seconds of voice speech). From this data, it is common in the speech

recognition community to extract the so-called Mel Frequency Cepstrum Coefficients (MFCCs) features (Reynolds et al., 2000). We selected $d = 40$ features for each speaker. Then, each speaker is modeled with a mixture of 16 different Gaussians. The test set consists of other 5 unseen utterances of the same 34 speakers (i.e. the training set and the test set have the same size). The task is to correctly label the unseen utterances with the name of the correct speaker. This is done by testing each of the GMM fitted during the training with the new voice sample, and selecting the model with the highest likelihood. Due to the differences in the speakers' audio data, the different dataset $V_1 \ldots V_{34}$ are made of a variable number of points which ranges from $n = 2000$ to $4000$.

In the vanilla configuration without thresholding, 169 among 170 utterances (5 utterances for all 34 speakers) were correctly labeled by EM with ML estimate, while all the elements in the test set were correctly recognized using the Bayesian (MAP) framework. During the training, we measured all the values of $\kappa(\Sigma), \kappa(V), \mu(V), \mu(\Sigma)$. For almost all GMM fitted (choosing a diagonal covariance matrix), there is at least a $\Sigma_j$ (among the 16 used to model a single speaker) with bad condition number (i.e up to 2500 circa). As in (Kerenidis et al., 2018; Kerenidis & Luongo, 2018) we took a threshold on the matrix by discarding singular values smaller than a certain value $m$. Practically, we discarded any singular value smaller than $2 \times 10^{-2}$. Thresholding the covariance matrices did not impact the accuracy (the generalization error) significantly. In the MAP estimate, only one element is not correctly classified, while this number goes up to two in the case of ML estimates. The results are in Table 1.

To check the resistence to noise, we perturbed each of the GMM $\gamma^t$, once fitted. Then, we measured variations of the accuracy on the test set. For each model, the perturbation consists of three things. First we add to $\theta$ a uniform vector (with randomly selected negative or positive entries) of $\ell_2$ norm of $\delta_\theta = 0.035$. Then we perturb each centroid $\mu_j$ with a vector of norm smaller than $\delta_\mu = 0.5$. In this error vector the sign of each noise component is chosen randomly, and the magnitude is sampled uniformly in the interval $(0, \frac{\delta_\mu}{\sqrt{d}})$. Then, we perturbed also the diagonal matrices $\Sigma_j$ with a vector of norm smaller than $\delta_\mu \sqrt{\eta}$, where $\eta = 10$. As we are using a diagonal GMM, this reduces to perturbing each singular value by some random noise upper bounded by $\pm \delta_\mu \sqrt{\eta}/\sqrt{d}$, making sure that each of the singular value stays positive, as covariance matrices are SPD. Last, the matrices are thresholded. Since the representation of the model used by our software stores the Cholesky's decomposition of the inverse, we worked with that representation. Notably, thresholding the $\Sigma_j$ help to mitigate the error of noise and regularize the model. With these parameters, we were able to correctly label 167 utterances over 170. We leave for the future further experiments with bigger and different types of datasets or where the noise is also added during the training process. We used scikit-learn (Pedregosa et al., 2011) to run all the experiments.

### 6.7 Quantum MAP estimate of GMM

Maximum Likelihood is not the only way to estimate the parameters of a model, and in certain cases might not even be the best one. For instance, in high-dimensional spaces, it is pretty common for ML estimates to overfit. Moreover, it is often the case that we have prior information on the distribution of the parameters, and we would like our models to take this information into account. These issues are often addressed using a Bayesian approach, i.e. by using a so-called Maximum A Posteriori estimate (MAP) of a model (Murphy, 2012, Section 14.4.2.8). MAP estimates work by assuming the existence of a *prior* distribution over the parameters $\gamma$. The posterior distribution we use as objective function to maximize comes from the Bayes' rule applied on the likelihood, which gives the posterior as a product of the likelihood and the prior, normalized by the evidence. More simply, we use the Bayes' rule on the likelihood function, as $p(\gamma; V) = \frac{p(V;\gamma)p(\gamma)}{p(V)}$. This allows us to treat the model $\gamma$ as a random variable, and derive from the ML estimate a MAP estimate:

$$\gamma_{MAP}^* = \arg\max_\gamma \sum_{i=1}^n \log p(\gamma|v_i) \tag{16}$$

Among the advantages of a MAP estimate over ML is that it avoids overfitting by having a kind of regularization effect on the model (Murphy, 2012, Section 6.5). Another feature consists in injecting into a maximum likelihood model some external information, perhaps from domain experts. This advantage comes at the cost of requiring "good" prior information on the problem, which might be non-trivial. In terms of labelling, a MAP estimates correspond to a *hard*

*clustering*, where the label of the point $v_i$ is decided according to the following rule:

$$y_i = \arg\max_j r_{ij} = \arg\max_j \log p(v_i|y_i = j; \gamma) + \log p(y_i = j; \gamma) \tag{17}$$

Deriving the previous expression is straightforward using the Bayes' rule, and by noting that the softmax is rank-preserving, and we can discard the denominator of $r_{ij}$ - since it does not depend on $\gamma$ - and it is shared among all the other responsibilities of the points $v_i$. Thus, from Equation 16 we can conveniently derive Equation 17 as a proxy for the label. Fitting a model with MAP estimate is commonly done via the EM algorithm as well. The Expectation step of EM remains unchanged, but the update rules of the Maximization step are slightly different. In this work we only discuss the GMM case, for the other cases the interested reader is encouraged to see the relevant literature. For GMM, the prior on the mixing weight is often modeled using the Dirichlet distribution, that is $\theta_j \sim \text{Dir}(\boldsymbol{\alpha})$. For the rest of parameters, we assume that the conjugate prior is of the form $p(\mu_j, \Sigma_j) = NIW(\mu_j, \Sigma_j|\boldsymbol{m}_0, \iota_0, \nu_0, \boldsymbol{S}_0)$, where $NIW(\mu_j, \Sigma_j)$ is the Normal-inverse-Wishart distribution. The probability density function of the NIW is the product between a multivariate normal $\phi(\mu|m_0, \frac{1}{\iota}\Sigma)$ and a inverse Wishart distribution $\mathcal{W}^{-1}(\Sigma|\boldsymbol{S}_0, \nu_0)$. NIW has as support vectors $\mu$ with mean $\mu_0$ and covariance matrices $\frac{1}{\iota}\Sigma$ where $\Sigma$ is a random variable with inverse Wishart distribution over positive definite matrices. NIW is often the distribution of choice in these cases, as is the conjugate prior of a multivariate normal distribution with unknown mean and covariance matrix. A shorthand notation, let's define $r_j = n\theta_j = \sum_{i=1}^n r_{ij}$. As in (Murphy, 2012), we also denote with $\overline{x_j}^{t+1}$ and $\overline{S_j}^{t+1}$ the Maximum Likelihood estimate of the parameters $(\mu_j^{t+1})_{ML}$ and $(\Sigma_j^{t+1})_{ML}$. For MAP, the update rules are the following:

$$\theta_j^{t+1} \leftarrow \frac{r_j + \alpha_j - 1}{n + \sum_j \alpha_j - k} \tag{18}$$

$$\mu_j^{t+1} \leftarrow \frac{r_j \overline{x_j}^{t+1} + \iota_0 \boldsymbol{m}_0}{r_j + \iota_0} \tag{19}$$

$$\Sigma_j^{t+1} \leftarrow \frac{\boldsymbol{S}_0 + \overline{S_j}^{t+1} + \frac{\iota_0 r_j}{\iota_0 + r_j}(\overline{x_j}^{t+1} - \boldsymbol{m}_0)(\overline{x_j}^{t+1} - \boldsymbol{m}_0)^T}{\nu_0 + r_k + d + 2} \tag{20}$$

Where the matrix $\boldsymbol{S}_0$ is defined as:

$$\boldsymbol{S}_0 := \frac{1}{k^{1/d}} Diag(s_1^2, \cdots, s_d^2), \tag{21}$$

where each value $s_j$ is computed as $s_j := \frac{1}{n}\sum_{i=1}^n (x_{ij} - \sum_{i=1}^n x_{ij}))^2$ which is the pooled variance for each of the dimension $j$. For more information on the advantages, disadvantages, and common choice of parameters of a MAP estimate, we refer the interested reader to (Murphy, 2012). Using the QEM algorithm to fit a MAP estimate is straightforward, since once the ML estimate of the parameter is recovered from the quantum procedures, the update rules can be computed classically.

**Corollary 6.20** (QEM for MAP estimates of GMM). *We assume we have quantum access to a GMM with parameters $\gamma^t$. For parameters $\delta_\theta, \delta_\mu, \epsilon_\tau > 0$, the running time of one iteration of the Quantum Maximum A Posteriori (QMAP) algorithm algorithm is*

$$O(T_\theta + T_\mu + T_\Sigma + T_\ell),$$

*for*

$$T_\theta = \widetilde{O}\left(k^{3.5}\eta^{1.5}\frac{\kappa^2(\Sigma)\mu(\Sigma)}{\delta_\theta^2}\right)$$

$$T_\mu = \widetilde{O}\left(\frac{kd\eta\kappa(V)(\mu(V) + k^{3.5}\eta^{1.5}\kappa^2(\Sigma)\mu(\Sigma))}{\delta_\mu^3}\right)$$

$$T_\Sigma = \widetilde{O}\left(\frac{kd^2\eta\kappa^2(V)(\mu(V') + \eta^2 k^{3.5}\kappa^2(\Sigma)\mu(\Sigma))}{\delta_\mu^3}\right)$$

$$T_\ell = \widetilde{O}\left(k^{1.5}\eta^{1.5}\frac{\kappa^2(\Sigma)\mu(\Sigma)}{\epsilon_\tau^2}\right)$$

*For the range of parameters of interest, the running time is dominated by $T_\Sigma$.*

