# OpenReview forum: "Quantum Expectation-Maximization for Gaussian Mixture Models"
_ICLR.cc/2020/Conference — Reject_

### Official Review · AnonReviewer1 · 2019-10-20
**Official Blind Review #1**

**Rating:** 3

**Review:**

Quantum machine learning is a hot topic, recently. There are several "schools of thought":
-- quantum kernel embeddings, which work per-observation, and allow for application on noisy intermediate-scale devices,
-- work based on quantum optimization solvers, which try to claim one could apply quantum eigensolvers to various training problems, but which does not seem to be applicable on intermediate-scale devices, given the scale of the requisite input,
--  novel quantum algorithms.
In principle, the novel quantum algorithms hold the promise of an exponential speed-up.

The authors propose a novel variant of expectation minimisation for parameter estimation of a gaussian mixture model with uniform mixing coefficients, targeting an ill-defined model of quantum computing of their own coinage.  Unfortunately, the paper is very sloppy.

The sloppiness starts with the model of quantum computing, which seems to assume anything the authors needed:
-- Definition 1: one can perform |k|d arithmetic operations in time polylog(d)
-- sentence above Lemma 3.4: post-selection (which makes it possible to solve at least all of PP, cf. https://arxiv.org/abs/quant-ph/0412187)
-- sentence below Lemma 3.6: "quantum linear algebra subroutines and tomography" in no time at all (?), wherein there are information-theoretic limits \Omega(exp(kd)) on the state tomography.

In terms of statistics and learning theory, the authors:
-- do not consider the separation of the Gaussians as a parameter. It is well-known that with o(1) separation, exponentially many samples are required, and with sqrt(log(k)) separation, polynomially many samples suggest (http://ieee-focs.org/FOCS-2017-Papers/3464a085.pdf). Unless Definition 1 of the authors "subsumes" processing of exponentially large numbers of samples in polylog(d) time, the authors may need to add an assumption on the separation.
-- the authors explain that their approach works only for the gaussian mixture model with uniform mixing coefficients only at the top of page 4, in a completely obscure notation of their own, and do not reference it as an assumption in the theorems later.

The sloppiness continues with the description of the Experiments in Section 4. Authors present a table of some results, but do not mention whether these have been obtained on quantum-computing hardware, a simulator thereof (what simulator? with noise?), or whether this is some fully-classical variant of the algorithm being tested. This clearly violates any "reproducibility checklist".

Within the "minor comments" category:
-- the introduction of the EM algorithms for GMM is sloppy. It is well known that EM for GMM is super-sensitive to noise and balance of the mixing coefficients (https://ieeexplore.ieee.org/document/8635825) and can get stuck in arbitrarily bad local optima, which is not mentioned once.
-- the references to other algorithms for GMM mention only Dasgupta 1999, rather than the subsequent 20 years of research, e.g., of Ankur Moitra at MIT (https://math.mit.edu/directory/profile.php?pid=1502).
-- on the other hand, there are plentiful references to arxiv pre-prints of Kerenidis et al, at least some of which have been shown to be vacuous, e.g., https://arxiv.org/abs/1808.09266 with an infinite upper bound on the run-time and no relation to the present paper?
-- there are some formulae missing or ending half-way through, e.g. Page 3 before "alas", Page 3 after "GMM would be".
-- there are a number of language issues: "we can thresholding", "some other real-world dataset".
-- the discussion starting with "Let's have a first high-level comparison" is completely wrong. Especially the condition number estimates of 5 seem to have no justification what so ever.
-- authors say that "polynomial dependence on the rank, the error, and the condition number, make these algorithms impractical on interesting datasets" -- but it is not clear whether they mean that their algorithm is also impractical?

Overall, while I like the idea of parameter estimation on a quantum computer, I could not recommend accepting the paper in its current form.

**Experience Assessment:**

I have published in this field for several years.

**Review Assessment: Checking Correctness Of Derivations And Theory:**

I carefully checked the derivations and theory.

**Review Assessment: Checking Correctness Of Experiments:**

N/A

**Review Assessment: Thoroughness In Paper Reading:**

I read the paper thoroughly.

---

> ### Author Response · Authors · 2019-11-12
> **Response to first part**
>
> Let us provide some facts in order to refute the reviewer’s opinion that we are using an ill-defined model of quantum computing. First, this is the usual model that assumes that the quantum algorithm has quantum access to the classical input of the problem, for example through a QRAM model. While we do not have experimental implementations of large scale QRAMs right now, similarly we do not have large scale quantum computers right now, the model is not ill-defined, in fact in the paper we should very clearly how quantum access to the parameters of the Gaussian Mixture Model enables us to perform a number of simple quantum operations that we use later on in the algorithms. Second, we of course do not assume post-selection in the sense of 0412187. In that paper, Post-selection means the ability to choose the outcome of a quantum measurement without having to pay the cost of having to repeat the measurement many times (inverse in the probability) or by performing amplitude amplification (which costs time inverse in the square root of the probability). In our case, we clearly state that we account for this cost in the running time, in other words we simply perform amplitude amplification in order to make the probability of the outcome of the measurement we want to achieve almost 1, thus selecting it. Third, when it comes to tomography, we state and expand on a tomography lemma that appeared in the q-means paper, that will appear in NeurIPS. The number of samples and running time of our procedure depends linearly on the dimension of the Hilbert space which is d and not exp(d). Just to make it even clearer, the state on which we are performing tomography has log(d) number of qubits and hence it takes time O(d). We repeat this procedure k times, therefore the runtime is O(kd).
>
>
> Separation of Gaussians
> Our work is providing a quantum analogue of the classical Expectation Maximization algorithm and inherits the strengths and weaknesses of the classical EM. Of course, more advanced algorithms for more Gaussian mixtures exist (not based on EM) and finding quantum analogues of those this is left for future work.
>
>
> Uniform mixing coefficients
> Probably the big-oh notation used in the dataset assumption section was not very clear, which lead to some confusion to the reviewer. In plain english, the assumption on the dataset can be stated as requiring that all the gaussian clusters have a proportional number of points, and not that the mixing coefficient are a uniform distribution. The assumption is used inside the proofs, which are moved in the supplementary material.
>
>
> Experiments
> Indeed while we have mentioned that these are results of simulations, we could have been even more clear to state that we have not used any quantum hardware (since no part of the algorithm can run using less than 50 qubits). We have simulated on a classical computer the quantum procedures, adding the corresponding errors on the results of the computations. We are happy to provide the code to run the experiments if the reviewer wishes us to do so.

---

> > ### Comment · AnonReviewer1 · 2019-11-13
> > **Another quick response**
> >
> > My first qualm is that the authors are not very upfront about the model of computing they use. If they said on page 1 "we utilise a highly speculative model of quantum computing, which involves QRAM and post-selection", that would actually be ok with myself. When they mention on page 6/8 that they, by the way, also require postselection etc, I find this at best sloppy -- and at worst deceptive.
> >
> > QRAM is definitely not part of any textbook definition of quantum computing, which would be either the quantum circuit or quantum TM model, although it has recently been popular in TCS conferences. I don't mind highly speculative models, if they allow for strong results.
> >
> > Post-selection is problematic in any case, though, because it makes it necessary to "step out" of the quantum circuit model and reason about some hybrid algorithms. The authors do not say "we present hybrid algorithms, featuring some quantum circuits and some classical computing", but rather make it seem it's all quantum. I find this at best sloppy -- and at worst deceptive.
> >
> > > Uniform mixing coefficients
> >
> > I did not say "uniformly distributed mixing coefficients", but "uniform mixing coefficients". That's what under some divisibility of the number of samples assumptions leads to the "proportional number of points", which you require in the proofs -- but do not mention in the statement in the theorem. Again, I would find that sloppy at best, and wrong at worst.

---

> > > ### Author Response · Authors · 2019-11-14
> > > **Final comment**
> > >
> > > In the analysis of the algorithm, we take into the account the computational cost for what we call post-selection, which is the process of taking an algorithm that outputs the correct outcome with some probability and making this probability go to 1 through amplitude amplification. This is NOT the notion of instantaneous Postselection as defined in Aaronson. In order to avoid confusion we will call it by a different name. Nonetheless, our model is not speculative nor unusual.
> > >
> > > For the QRAM, we clearly state that we need it in the main Result of the paper in page 2.
> > >
> > > The uniform mixing coefficients assumption allows us to give better bounds for the running time.
> > > The algorithm does not require this assumption, it appears only in the analysis. We will clearly state in future versions when this assumption when it is being used.
> > >
> > >
> > > The results of experiments for the range of the condition number are given in table 1. We found the average condition number can be close to 5, and the range for the condition numbers to be [1,50] for these experiments. The one data point in the text is a typical sample, we will clarify this discussion in future versions and refer to Table 1.
> > >
> > > Regarding the citations to Kerenidis et al. IPM paper, we use a tomography algorithm from this paper, this is clearly stated in Theorem 6.10.
> > >
> > > Regarding the referees views about the vacuity of the above paper, it seems that approximation algorithms which produce epsilon approximate solutions and have running time scaling 1/epsilon would be considered vacuous by the referee as they require infinite time when epsilon goes to 0. Such a view would render many well known numerical algorithms vacuous and would leave the referee with a rather large vacuum to fill.
> > >
> > > A more legitimate question is on the growth rate of the condition number with respect to epsilon, from our discussions with practitioners in optimization suggest that empirically kappa = O(1/epsilon), though rigorous results along these lines are lacking.

---

> ### Author Response · Authors · 2019-11-12
> **Response to second part**
>
>
>
> For the "minor comments":
> We briefly introduced EM using the notation from Andrew Ng’s Stanford lectures, but we can give more details space permitting. We talk in many places in the paper about “local” minima, as the solutions found by the EM algorithm. Again our algorithm inherits the same properties of the EM algorithm, and we believe that the importance of the classical EM algorithm for unsupervised learning is self-evident.
>
> We will add references to the classical EM literature.
>
> We are not sure how scientific or constructive it is to call a work “vacuous”, especially a work which provides the first quantum interior point method for solving Semi-definite programming with a running time that also depends on the condition number of the matrices. Many classical methods also depend on the condition number of the matrices and this indeed means that one can construct matrices for which the real condition number is very large (not sure what infinite means as the analysis of these methods proves that the matrices involved are invertible). This does not make any of these methods vacuous, it just means that one needs to be careful when applying them and needing to take care to either precondition the matrices or use a threshold instead of the real condition number in order to make the algorithm work even for the cases of ill-conditioned matrices. Note also that the HHL algorithm also depends on the condition number. The referee may want to argue that even that algorithm is vacuous but we would like to clearly state that this is an extremely personal view of the referee which certainly does not agree with the scientific community in this domain that has been producing quantum algorithms with a dependence on the condition number for quite a few years now. Moreover, we go one step further and we try to analyze what the condition number actually is for certain data sets that appear in practice. In the q-means paper, an analysis of the MNIST data set appears with the condition numbers of the matrices, which not only are they not “infinite”, rather they stay constant with increasing dimension.
>
> The referee also claims that the paragraph "Let's have a first high-level comparison" is “completely wrong”. The only reason put forward is that we estimate the condition number to be 5 without any justification. In fact, our justification is very clear: the condition number comes from the experiment section.  We thus fail to understand the “completely wrong” characterization of the referee.
>
> With respect to the quantum-inspired algorithms, we claim that their dependence on the rank, the error, and the condition number, make these algorithms impractical on interesting datasets”, because this dependence is to the power of at least 18 or even 88 for some cases. So indeed they depend polynomially in these parameters but in a way that renders them highly impractical. The dependence of our quantum algorithms on these parameters are linear or very small polynomials, which make our algorithms better suited to be practical. For this one would of course need to implement them on real quantum computers and test their behavior, but given the absence of such quantum computers, the best we can hope is a rigorous mathematical analysis and simulation results on some small data sets.

---

> > ### Comment · AnonReviewer1 · 2019-11-13
> > **A Quick Response**
> >
> > I do not plan to engage in an extensive discussion of papers by Kerenidis et al. My point was twofold:
> > -- that there are references to papers of Kerenidis, which seem to have very little to do with the parameter estimation in Gaussian Mixture Models. (SDP IPM on quantum computers? Can you formulate the parameter estimation as an SDP? If so, that would make the reference legit, maybe.)
> > -- some of those papers, such as the SDP IPM on quantum computers, feature poor analysis. The paper on SDP IPM actually parametrised the run-time by condition number of the KKT system. It is well known that as the complementary slackness goes to zero, the condition number of the KKT system goes to infinity. If an upper bound on run-time is "infinity", I'd call it vacuous.
> >
> > On the "5 without any justification" issue: the original paper says "It is crucial to find datasets where such a quantum algorithm can offer a speedup. For a reasonable range of parameters". Then, it proceeds to give a single point in the parameter space, which involves condition number 5. I don't think that assuming condition number 5 is "reasonable" -- or "range", for that matter. If the authors said condition numbers in [1, 100,000] are a "reasonable range", I would be ok. This way, the paragraph tries to make the impression that the authors have found a "reasonable range", where there is speed-up, which I could not agree with. Perhaps the authors could have said "we are looking for a reasonable range of parameters. Our simulations use X, Y, Z".

---

### Official Review · AnonReviewer3 · 2019-10-24
**Official Blind Review #3**

**Rating:** 3

**Review:**

The authors present a quantum algorithm for expectation maximization on a quantum machine, which is poly-logarithmic in the size of the dataset (and polynomial in other parameters such as the dimension of the feature space, and number of mixture components, condition number of the data and covariance matrices, some precision/error parameters etc) per iteration. So, compared to the . regular EM algorithm, this yields exponential speedup in the number of data points, but is worse in other factors (such as a k^4.5 dependence on the number of mixture components). So, the quantum system could be superior to a conventional computer for some settings of parameters. They run some (simulation) experiments on a dataset (VoxForge) to report accuracies (though there is no comparison of the accuracy of the classical algorithm). Also, there does not seem to be any experimental results to confirm the theoretical analysis of the scaling characteristics (i.e., to show that the scaling is as predicted by the theory).


**Experience Assessment:**

I do not know much about this area.

**Review Assessment: Checking Correctness Of Derivations And Theory:**

I assessed the sensibility of the derivations and theory.

**Review Assessment: Checking Correctness Of Experiments:**

I assessed the sensibility of the experiments.

**Review Assessment: Thoroughness In Paper Reading:**

I read the paper at least twice and used my best judgement in assessing the paper.

---

### Official Review · AnonReviewer2 · 2019-11-05
**Official Blind Review #2**

**Rating:** 1

**Review:**

The authors present and analyze a quantum computing algorithm for learning GMMs.

I think this paper cannot be accepted because it violates formatting guidelines. Also, I think it is not appropriate for ICLR since it assumes knowledge of quantum computing that most people at this conference would not have, and I as a reviewer do not possess, and hence cannot evaluate this paper. For example, I do not know bra-ket notation.

If the ACs disagree, I am happy to revise my review for this paper and try to be more thorough.

**Experience Assessment:**

I do not know much about this area.

**Review Assessment: Checking Correctness Of Derivations And Theory:**

I did not assess the derivations or theory.

**Review Assessment: Checking Correctness Of Experiments:**

I did not assess the experiments.

**Review Assessment: Thoroughness In Paper Reading:**

I made a quick assessment of this paper.

---

> ### Author Response · Authors · 2019-11-07
> **Please add more explainations**
>
> Hello,
> Thanks for your comments.
> While we are well-aware that ICLR is a venue for classical machine learning algorithm, we do have witnessed an increase in the number of quantum machine learning papers proposed there.
>
> For instance, a quick search on the current openreview website for ICLR2020 shows 9 other quantum-related paper, among which some of them are quantum algorithms.  We also would like to point out that at NeurIPS2019 a similar paper which we generalized (q-means) has been accepted. Other notable quantum papers accepted at classical ML conferences are Quantum Perceptron Models at NIPS 2016, and Online Learning of quantum states,  NeurIPS 2018.
>
> For what it concerns the formatting guidelines, can you please be more precise? We used the template and cannot find any formatting violations (i.e. for instance we have exactly 8 pages of main text, as suggested).  We are certainly open to patch any mistake before the final submission.
>
> Thanks.

---

> > ### Comment · AnonReviewer2 · 2019-11-08
> > **Formatting**
> >
> > I'm fairly certain that your margins are significantly narrower relative to the official ICLR margins. Just compare to any recent ICLR paper. I don't know how strict the ACs are about this, but traditionally this has been a hard rule (otherwise, it's unfair to other authors that have less space to present their work).
> >
> > In terms of quantum ML papers, that's fair, but I am just not qualified to review quantum computing papers, especially if there is not enough background on basic notation.

---

### Decision · Program_Chairs · 2019-12-19

**Decision:**

Reject

**Comment:**

The reviewers were unanimous that this submission is not ready for publication at ICLR in its current form.

Concerns raised include a significant lack of clarity, and the paper not being self-contained.